# Lupus enhancer risk variant causes dysregulation of *IRF8* through cooperative lncRNA and DNA methylation machinery

Tian Zhou[1,2,3,16], Xinyi Zhu[1,16], Zhizhong Ye[3,16], Yong-Fei Wang [4], Chao Yao[5], Ning Xu[1], Mi Zhou [6], Jianyang Ma[1], Yuting Qin[1], Yiwei Shen[1], Yuanjia Tang[1], Zhihua Yin[3], Hong Xu[7,8], Yutong Zhang[1], Xiaoli Zang[1], Huihua Ding[1], Wanling Yang [4], Ya Guo[6], John B. Harley[9], Bahram Namjou [10], Kenneth M. Kaufman[10,11,12], Leah C. Kottyan [10,12,13], Matthew T. Weirauch [10,12,14,15], Guojun Hou [1,2,3✉] & Nan Shen [1,2,3,10,12✉]

Despite strong evidence that human genetic variants affect the expression of many key transcription factors involved in autoimmune diseases, establishing biological links between non-coding risk variants and the gene targets they regulate remains a considerable challenge. Here, we combine genetic, epigenomic, and CRISPR activation approaches to screen for functional variants that regulate *IRF8* expression. We demonstrate that the locus containing rs2280381 is a cell-type-specific enhancer for *IRF8* that spatially interacts with the *IRF8* promoter. Further, rs2280381 mediates *IRF8* expression through enhancer RNA AC092723.1, which recruits TET1 to the *IRF8* promoter regulating *IRF8* expression by affecting methylation levels. The alleles of rs2280381 modulate PU.1 binding and chromatin state to regulate AC092723.1 and *IRF8* expression differentially. Our work illustrates an integrative strategy to define functional genetic variants that regulate the expression of critical genes in autoimmune diseases and decipher the mechanisms underlying the dysregulation of *IRF8* expression mediated by lupus risk variants.

[1] Shanghai Institute of Rheumatology, Renji Hospital, Shanghai Jiao Tong University School of Medicine (SJTUSM), Shanghai 200001, China. [2] State Key Laboratory of Oncogenes and Related Genes, Shanghai Cancer Institute, Renji Hospital, Shanghai Jiao Tong University School of Medicine (SJTUSM), Shanghai 200032, China. [3] Shenzhen Futian Hospital for Rheumatic Diseases, Shenzhen 518040, China. [4] Department of Paediatrics and Adolescent Medicine, The University of Hong Kong, Hong Kong 999077, China. [5] Shanghai Institute of Nutrition and Health, Shanghai Institutes for Biological Sciences (SIBS), University of Chinese Academy of Sciences, Chinese Academy of Sciences (CAS), Shanghai 200031, China. [6] Sheng Yushou Center of Cell Biology and Immunology, Joint International Research Laboratory of Metabolic and Developmental Sciences, School of Life Sciences and Biotechnology, Shanghai Jiao Tong University (SJTU), Shanghai 200240, China. [7] Department of Obstetrics and Gynecology, Renji Hospital, Shanghai Jiao Tong University School of Medicine (SJTUSM), Shanghai 200127, China. [8] Shanghai Key Laboratory of Gynecologic Oncology, Renji Hospital, Shanghai Jiao Tong University School of Medicine (SJTUSM), Shanghai 200127, China. [9] US Department of Veterans Affairs Medical Center, Cincinnati, OH 45229, USA. [10] Center for Autoimmune Genomics and Etiology, Cincinnati Children's Hospital Medical Center, Cincinnati, OH 45229, USA. [11] Division of Immunobiology, Cincinnati Children's Hospital Medical Center, Cincinnati, OH 45229, USA. [12] Department of Pediatrics, University of Cincinnati College of Medicine, Cincinnati, OH 45229, USA. [13] Division of Allergy and Immunology, Cincinnati Children's Hospital Medical Center, Cincinnati, OH 45229, USA. [14] Division of Developmental Biology, Cincinnati Children's Hospital Medical Center, Cincinnati, OH 45229, USA. [15] Division of Biomedical Informatics, Cincinnati Children's Hospital Medical Center, Cincinnati, OH 45229, USA. [16]These authors contributed equally: Tian Zhou, Xinyi Zhu, Zhizhong Ye. ✉email: houguojun225@163.com; nanshensibs@gmail.com

Transcription factors (TFs) are specialized proteins that bind to specific sequences of DNA and activate or repress the transcription of genes[1]. The use of TFs knockout mice, such as *IRF5*, has directly revealed the critical role of TFs in the pathogenesis and severity of autoimmune diseases[2]. Furthermore, inhibition of TFs can effectively intervene in disease progression, making TFs attractive therapeutic targets in many diseases[3–5].

The expression levels of many genes vary from person to person, and genetic variants may make a significant contribution[6–8]. In particular, genetic variants can alter the expression of genes encoding TFs, resulting in altered expression levels of downstream genes controlled by specific TFs[9–11]. For example, *BCL11A* plays a crucial role in suppressing γ-globin expression and fetal hemoglobin in erythrocytes. Genome-wide association studies (GWAS) have found that genetic variants at the *BCL11A* locus are associated with fetal hemoglobin expression levels, and targeting *BCL11A* could prevent or ameliorate complications of sickle cell disease by regulating γ-globin expression levels[11,12]. Therefore, dissecting the effects of functional genetic variants on TF expression could help elucidate the aberrant expression of TFs in disease mechanisms, especially for diseases with genetic susceptibility.

Autoimmune diseases are complex and heterogeneous diseases primarily caused by immune responses directed against oneself[13]. Many autoimmune diseases, such as systemic lupus erythematosus (SLE), have genetic predisposition[14–16]. Aberrant expression of TFs leads to dysregulation of multiple signaling pathways. It is thought to contribute extensively to autoimmune disease[17,18]. However, few autoimmune diseases risk genetic variants are directly associated with TF expression levels. This is because the genetic variants identified by GWAS are not necessarily causal due to linkage disequilibrium (LD)[14]. In addition, most variants are in non-coding genomic regions, and are therefore more likely to function in a cellular or context-dependent manner[19–22]. Furthermore, disease risk genes are usually defined based on genomic proximity or expression quantitative trait loci (eQTL) signatures, which do not necessarily identify disease-causing genes.

We propose a strategy to decipher the genetic regulatory mechanisms of TF expression in diseases using *IRF8* as an example to fill this gap. Several genetic studies have nominated *IRF8* as an important autoimmune disease risk gene[23–30]. Consistent with this notion, the function of *IRF8* is associated with a variety of autoimmune-related phenotypes, such as immune cell development, inflammatory cytokine production, regulation of IFN-stimulated gene (ISG) expression[31–33]. Despite the substantial heritability of the *IRF8* locus, the functional variants, causative genes, and underlying gene regulatory mechanisms involved in autoimmune disease are largely unknown. Here, we combine genetic data, epigenomic analysis, CRISPR activation (CRISPRa) screen, CRISPR-mediated knockout and 3D chromatin structure analysis to identify functional variants in the *IRF8* locus. We demonstrate that rs2280381 is likely a causal variant that regulates *IRF8* expression through modulating enhancer RNA (eRNA) expression and cell-type-specific enhancer-promoter loop interactions. In addition, eRNA interacts with TET1, which binds to the *IRF8* promoter and regulates its methylation levels to regulate *IRF8* expression. In particular, the rs2280381 allele differentially affects TF occupancy and chromatin status to fine-tune *IRF8* expression, thereby contributing to disease pathogenesis.

## Results

**CRISPR activation screen identifies the functional autoimmune diseases associated genetic variants at the *IRF8* locus.** *IRF8* locus is strongly associated with autoimmune diseases, including Bechet's disease[30], rheumatoid arthritis[34,35], systemic sclerosis[36,37], SLE[24,25,28,38] and multiple sclerosis (MS)[23,39,40]. GWAS have reported at least 16 genetic variants that are genome-wide significantly associated with autoimmune disease in this locus[23–25,28,30,34–40]. However, the functional variants are currently unknown. To prioritize autoimmune diseases risk variants with potential regulatory function on *IRF8* expression, we developed a strategy to screen the genetic variants with CRISPRa assay using gRNAs targeting the SNP-containing region. We first collected all autoimmune disease-associated genetic variants with genome-wide significance ($P < 5 \times 10^{-8}$) published through 2020[23–25,28,30,34–40] (Fig. 1A and Supplementary Data 1). To include all possible disease-associated variants, we further included all SNPs in tight LD ($r^2 > 0.8$) with these tag variants according to the reported population, and identified 89 SNPs in total (Supplementary Data 2). Since most GWAS variants are located in non-coding regions of the genome, and variants impacting gene regulation are often located within enhancer regions, we analyzed active enhancer signals including H3K27ac modification and chromatin accessibility (ATAC-seq and DNase I hypersensitive sites (DHSs)) of the above SNPs in four major human immune cell subpopulations. SNPs with high H3K27ac, DNase-seq, or ATAC-seq signal in any immune cell subpopulation were considered as candidates (Fig. 1B). This procedure identified 32 candidate genetic variants (Supplementary Data 3). Among these SNPs, 18 SNPs are located in monocyte-specific enhancers, with the remaining SNPs mostly occurring in shared enhancer regions of CD4+ T cells, CD8+ T cells, CD19+ B cells and CD14+ Monocytes (Supplementary Data 3 and Fig. 1C). Interestingly, nearly all of these candidate SNP-containing regions are enhancers in CD14+ monocytes (Supplementary Data 3 and Fig. 1C). Based on the above observations, we decided to perform our functional screen assays in monocytes.

To begin functionally identifying the regulatory potential of these SNPs, we first established cell line stably expressing dCas9-VP64 in U-937 monocyte cells. Three gRNAs around each SNP were designed and synthesized. The gRNA mixture was then transfected into the cells for 24 h and *IRF8* mRNA expression levels were measured (Fig. 1B). The results suggested that 12 genomic regions containing variants could induce greater than 1.4-fold increases in *IRF8* expression levels (Fig. 1D). To confirm this result, we further carried out CRISPR SAM activation assay[41] and KRAB-dCas9 interference[42] assay to validate the regulatory effect of top 5 candidate regions. As shown in Supplementary Fig. 1, these regions demonstrate the strong regulatory effect on *IRF8* expression in both assays, with the SLE risk SNP rs2280381-containing region having the strongest regulatory effect among these SNPs. Moreover, rs2280381 is located within a monocyte-specific enhancer (Fig. 1E). Based on these results, we focused our study on rs2280381 and SLE.

**The rs2280381-containing enhancer regulates *IRF8* expression in a cell-type-dependent manner via enhancer-promoter connections.** CRISPR/Cas9-mediated deletion is a widely used tool to study enhancer function. To directly assess the regulatory function of the rs2280381-containing region, we generated cell clones with an ~138-bp deletion at the rs2280381 locus using CRISPR/Cas9 technology in U-937 cells (Fig. 2A, B). These clones underwent the same procedure but with the wildtype genotype as a negative control. As expected, deletion of the fragment containing rs2280381 resulted in a significant reduction in *IRF8* expression, both at the mRNA and protein level (Fig. 2C, D). In addition, we examined the enhancer-tagged signal in this region by analyzing the publicly available H3K27ac ChIP-seq data and ATAC-seq data in U-937 cells[43]. We found that this chromatin

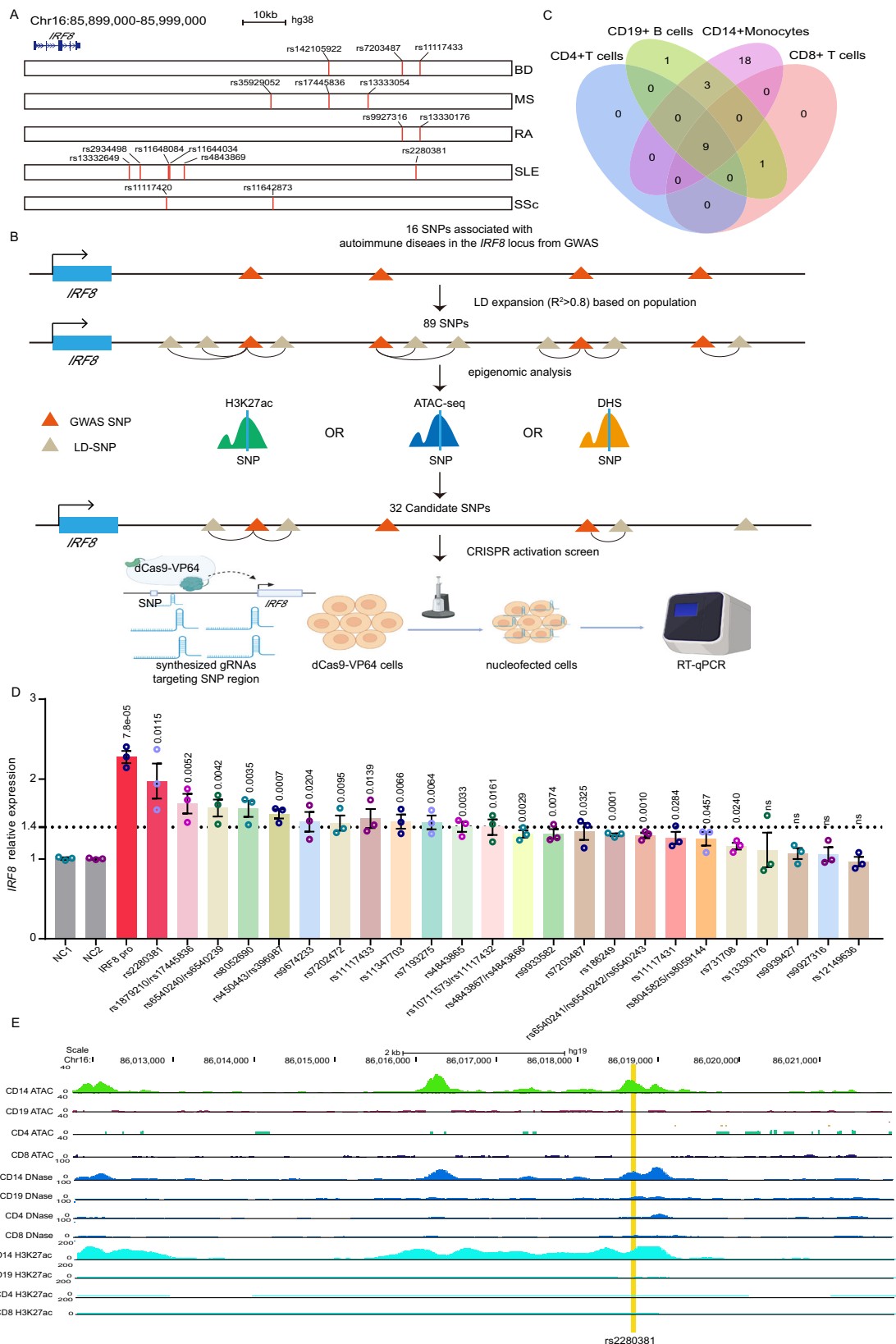

region was open and highly modified by the H3K27ac marker (Fig. 2E), further confirmed by FAIRE-qPCR and ChIP-qPCR (Supplementary Fig. 2A, B). Together, these data suggest that the region containing rs2280381 is a functional enhancer that regulates *IRF8* expression in U-937 cells.

Distal enhancers typically form enhancer-promoter loops that affect the expression of target genes. To test whether such a linkage exists between the *IRF8* promoter and the rs2280381 enhancer, we performed circularized chromosome conformation capture sequencing (4C-seq) to detect looping interactions of the

**Fig. 1 CRISPR activation screen reveals functional genetic variants modulating *IRF8* expression. A** The position of GWAS tag SNPs (shared *x*-axis indicated above) with respect to the *IRF8* gene for autoimmune diseases (*y*-axis). **B** Strategy for choosing candidate autoimmune disease-associated SNPs for the CRISPR activation screen. DHS DNase I hypersensitive site. **C** The Venn diagram indicates the overlap of SNPs with enhancer signals among different human immune cell subpopulations. **D** RT-qPCR analysis of *IRF8* expression in the CRISPR activation experiment (*n* = 3, biologically independent experiments). **E** Chromatin landscape analysis reveals that rs2280381 is located within a likely cell-type-specific enhancer. BD Behçet's disease, MS multiple sclerosis, RA rheumatoid arthritis, SLE systemic lupus erythematosus, SSc systemic sclerosis, GWAS genome-wide association studies, LD linkage disequilibrium. Data are represented as mean ± SEM, and *p* values are calculated using an unpaired two-tailed Student's *t* test. ns not significant. See also Supplementary Fig. 1.

*IRF8* promoter within this region. This assay revealed a circular physical interaction between the rs2280381 enhancer and the *IRF8* promoter (Fig. 2F). Furthermore, this observation was further confirmed based on the rs2280381 view point (Fig. 2G). Consistent with these observations in U-937 cells, we also observed communication between the *IRF8* promoter and the rs2280381 locus in primary monocytes (Supplementary Fig. 2C).

Since enhancers are usually cell-type-specific, and our data suggest that the rs2280381 enhancer is a monocyte-specific enhancer, we next sought to define in which cell type this region has a regulatory function. To this end, we first deleted the region containing rs2280381 in Raji (B cell) cells and Jurkat (T cell) cells, and found that deletion of this region did not affect *IRF8* expression (Fig. 2H, I). Next, we isolated CD14+ monocytes, CD3+ T cells and CD19+ B cells from human peripheral blood mononuclear cells (PBMCs) and disrupted the rs2280381 region by delivering Cas9 RNP to these cells. After editing, cells were collected to extract RNA and genomic DNA (Supplementary Fig. 2D). The editing effect was estimated using ICE (https://ice.synthego.com/#/). Gene expression was selected for examination by analyzing Sanger sequencing results of target motifs and samples with efficiency higher than 30% (Supplementary Figs. 3–5). As shown in Fig. 2J–L, the disruption of the rs2280381-containing region only affected *IRF8* expression in monocytes. This is consistent with the epigenetic modifications (Fig. 1E) observed in these immune cells. In conclusion, these data suggest that the genomic region containing rs2280381 is a cell-type-specific enhancer, forming an enhancer-promoter interaction to regulate the expression of *IRF8*.

**LncRNA AC092723.1 near rs2280381 acts as an enhancer RNA to regulate *IRF8* expression.** To further identify downstream targets that the rs2280381-containing region may regulate, we performed RNA-seq on three WT and three KO clones. We then performed differential gene expression analysis and identified 149 and 247 genes that were significantly down- and upregulated by deletion of this region, respectively (log2 fold-change ≥1 and false discovery rate (FDR) cutoff <0.05) (Supplementary Fig. 6A and Supplementary Data 4). Enhancers usually engage in chromosomal interactions with their target promoters to regulate gene expression. To explore genes directly modulated by the rs2280381-containing region, we first analyzed the 4C-seq data from the perspective of rs2280381, and found that the promoter regions of 66 genes interacted with the rs2280381 site (Supplementary Data 5). Comparing the expression of these genes between WT and rs2280381 KO cell clones, we found that the expression levels of lncRNA AC092723.1 and *IRF8* were significantly different between the two groups (log2 fold-change ≥1 and FDR < 0.05) (Supplementary Data 6), suggesting AC092723.1 and *IRF8* may be affected by direct regulation of the rs2280381-containing enhancer region. After demonstrating that *IRF8* as a direct target of the rs2280381-containing region, we used the Ingenuity Pathway Analysis to predict the possible targets of *IRF8*. In parallel, IRF8 ChIP-seq data in monocytes (http://cistrome.org/db/#/)[44,45] was analyzed to identify gene promoter

region enriched for IRF8 binding, which we considered direct *IRF8* target. These analyses yielded 61 gene candidates (Supplementary Data 7). For some genes, such as *ISG15*, *IFIT1* and *CIITA*, we performed RT-qPCR to verify their expression. Consistent with the previous reports, *IRF8* could negatively regulate the expression of *ISG15*[46] and *IFIT1*[32] and positively regulate the expression of *CIITA*[47] (Supplementary Fig. 6B–D). Next, differentially expressed genes were analyzed using gene ontology and found highly enriched in biological process such as inflammatory response, response to LPS, innate immune response, macrophage activation (Supplementary Fig. 6E and Supplementary Data 8), which coincides with the established functions of *IRF8*[31].

Analysis of RNA-sequencing data indicated that the expression of AC092723.1 and *IRF8* was regulated by the rs2280381-containing region (Fig. 3A), an observation was further validated by RT-qPCR (Fig. 3B). AC092723.1 is located upstream of rs2280381, and the distance between the 3′ end of AC092723.1 and rs2280381 is ~300 bp (Fig. 3C). Epigenomic analysis revealed that the genomic region of AC092723.1 overlaps with an extensive monocyte-specific possible enhancer with strong H3K4me1, H3K27ac, ATAC and DHS signaling (Supplementary Fig. 6F). Targeting CRISPR SAM to the rs2280381-containing region strongly upregulated both AC092723.1 and *IRF8* expression (Fig. 3D, E). Based on these observations, we hypothesized that AC092723.1 may act as an eRNA that mediates the regulation of *IRF8* expression by rs2280381.

Since lncRNA expression levels are usually tissue- or cell-specific, we first investigated the abundance of AC092723.1 in different human immune cell subpopulations. Consistent with the chromatin landscape of this region, AC092723.1 was highly expressed in human CD14+ monocytes (Fig. 3F). This was also validated by public RNA-sequencing data from different immune cell subpopulations (Supplementary Fig. 6G). In addition, we examined the intracellular localization of AC092723.1 in primary monocytes and U-937 cells by cell fractionation and RT-qPCR. We observed that AC092723.1 was predominantly distributed in the nuclear fraction (Fig. 3G and Supplementary Fig. 6H), similar to most regulatory lncRNAs. To directly assess the regulatory function of AC092723.1, we knocked down this lncRNA by antisense oligonucleotides (ASOs) and tested the expression of *IRF8*. As shown in Fig. 3H, I and Supplementary Fig. 6I, J, knockdown of AC092723.1 significantly reduced *IRF8* expression in both primary monocytes (Fig. 3H, I) and U-937 cells (Supplementary Fig. 6I, J). In contrast, knockdown of *IRF8* with siRNA did not reduce the expression of AC092723.1 (Fig. 3J, K and Supplementary Fig. 6K, L). We further confirmed this result by deleting part of the AC092723.1 region in U-937 cells with CRISPR/Cas9-mediated fragment deletion (Supplementary Fig. 6M and Fig. 3L). In conclusion, these data provide direct evidence that the rs2280381 enhancer governs the expression of eRNA AC092723.1 to regulate the expression of *IRF8*.

**AC092723.1 interacts with the TET1 protein and binds to the *IRF8* promoter to regulate *IRF8* expression by affecting methylation levels.** To explore the mechanism by which

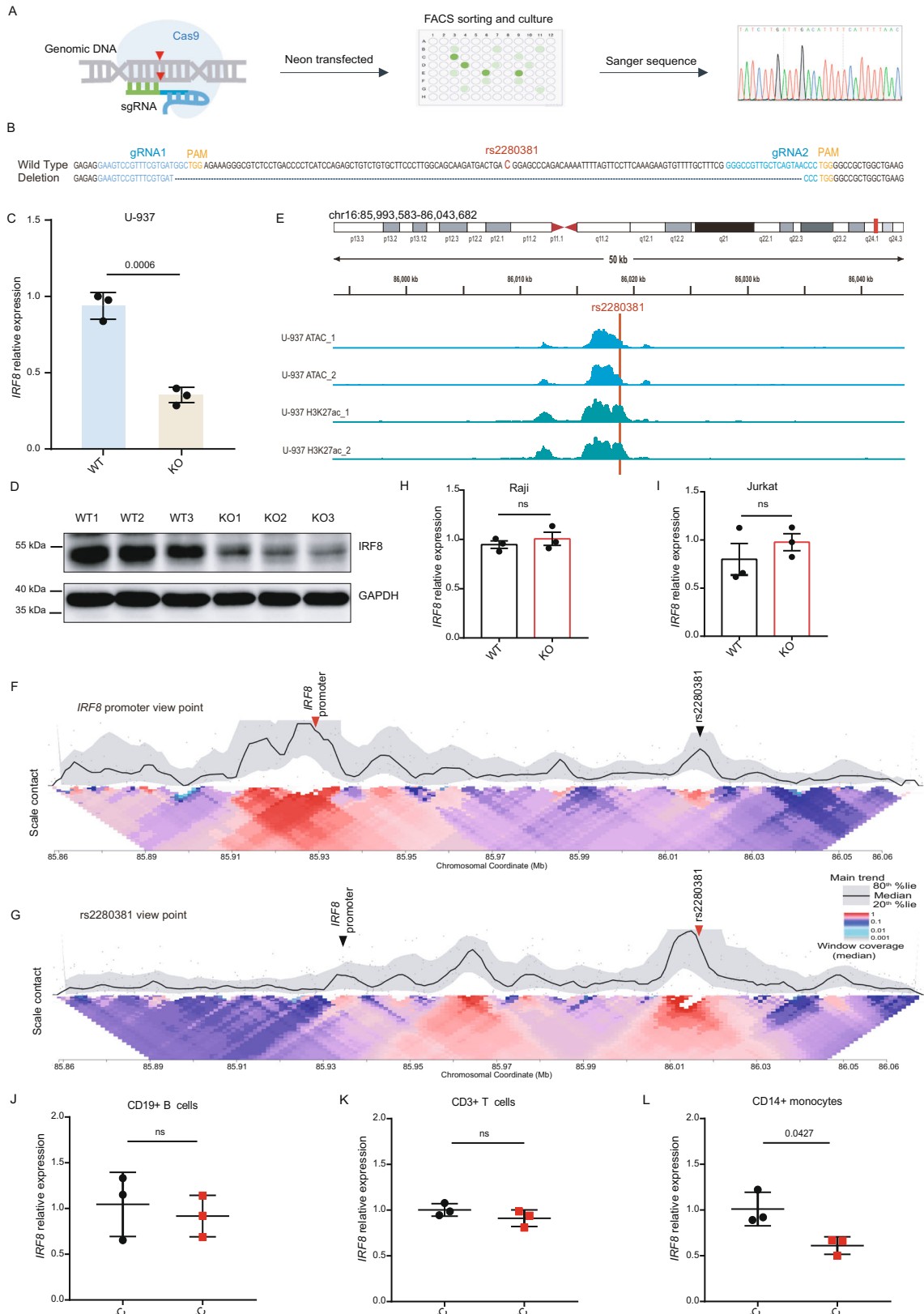

AC092723.1 *cis*-regulates *IRF8* expression, we first performed a chromatin isolation by RNA purification (ChIRP) experiment[48] to assess the interaction between lncRNA and the *IRF8* promoter. We designed biotinylated ASOs and incubated these probes with chromatin fractions from primary monocytes or U-937 cells. The core binding sequence of the *IRF8* promoter was assessed by RT-qPCR (Fig. 4A). Five pairs of PCR primers were designed relative to the *IRF8* transcription start site from −1000 to +153, including all regions with high chromatin accessibility assessed by ATAC-seq (Supplementary Fig. 7A). The AC092723.1 probe was firmly and enriched explicitly for AC092723.1 RNA compared to the control NEAT1 (Fig. 4B, C and Supplementary Fig. 7B, C).

**Fig. 2 The rs2280381-containing region is a cell-type-dependent enhancer regulating *IRF8* expression. A** Flow chart for generating genomic fragment deletion clones using the CRISPR-Cas9 technology. **B** The genotype of rs2280381 wildtype clones and deletion clones. **C** RT-qPCR analysis of *IRF8* expression in U-937 WT and KO clones (*n* = 3, biologically independent samples). **D** WB analysis of *IRF8* expression in U-937 WT and KO clones. Blot is representative of *n* = 3 biologically independent experiments. **E** Analysis of active enhancer signals (chromatin accessibility and H3K27ac) within the rs2280381-containing region in U-937 cells by ATAC-seq and ChIP-seq. 4C-seq analysis of contact profiles of the *IRF8* promoter (**F**) and the rs2280381 site (**G**) using a 2 kb window size in the main trend subpanel. Red arrowhead indicates the view point position, and black arrowhead indicates the target position. Gray dots indicate normalized contact intensities. The heat map displays a set of medians of normalized contact intensities calculated at different window sizes. RT-qPCR analysis of *IRF8* expression in Raji (**H**) or Jurkat (**I**) WT and KO clones (*n* = 3, biologically independent samples). RT-qPCR analysis of *IRF8* expression in CRISPR/Cas9 RNP edited CD19+ B cells (**J**), CD3+ T cells (**K**) and CD14+ monocytes (**L**) (*n* = 3, biologically independent samples). WT: rs2280381 wildtype, KO: 138 bp fragment containing the rs2280381 deletion. Data are represented as mean ± SEM, and *p* values are calculated using an unpaired two-tailed Student's *t* test (**C**, **H**, **I**) and paired two-tailed Student's *t* test (**J–L**). ns not significant. See also Supplementary Figs. 2–5.

More importantly, analysis of the DNA sequences pulled down by the AC092723.1 probe showed significant enrichment in the −473/−395 *IRF8* promoter sequence (Fig. 4D and Supplementary Fig. 7D). In addition, we also found that AC092723.1 could interact with the region containing rs2280381 (Fig. 4E and Supplementary Fig. 7E), suggesting that AC092723.1 may contribute to the formation of a loop between the *IRF8* promoter and the rs2280381 enhancer (Fig. 4F). Consistent with this observation, knocking down AC092723.1 by ASO significantly decreased the interaction frequency between the *IRF8* promoter and rs2280381 (Fig. 4G, H).

Recent studies have demonstrated that many lncRNAs can act as scaffolds for chromatin-modifying enzymes, regulating chromatin epigenetic modifications to enhance or repress gene expression[49,50]. To test whether AC092723.1 can interact with epigenetic modifying enzymes, we first used the RPISeq (http://pridb.gdcb.iastate.edu/RPISeq/references.php)[51,52] to predict possible binding epigenetic modifying enzymes with this lncRNA. Since AC092723.1 positively regulated the expression of *IRF8*, we focused our candidates on chromatin modifiers with potential function to activate gene expressions, such as WDR5, EP300, TET1 and DOT1L (Supplementary Fig. 7F). We performed an RNA-binding protein immunoprecipitation assays (RIP) with antibodies specific for each of the above chromatin modifiers to test these candidate modifiers. As shown in Supplementary Fig. 7G, only the anti-TET1 antibody was enriched in high abundance for AC092723.1 relative to the IgG control. In addition, the binding between TET1 and AC092723.1 was confirmed in primary monocytes (Fig. 4I).

Next, we investigated whether TET1 plays a functional role with AC092723.1 in regulating the expression of *IRF8*. We knocked down *TET1* with siRNA in primary monocytes or U-937 cells and performed RT-qPCR analysis to assess knockdown efficiency and *IRF8* expression levels. The results showed that silencing of *TET1* significantly down-regulated *IRF8* expression (Fig. 4J, K and Supplementary Fig. 7H, I). Furthermore, to elucidate how the AC092723.1-TET1 complex regulates *IRF8* expression, we analyzed the enrichment of TET1 in the *IRF8* promoter region by performing chromatin immunoprecipitation (ChIP) assay in primary monocytes and U-937 cells, with or without lncRNA knockdown. The results showed that TET1 could directly bind to the *IRF8* promoter region. This binding activity was impaired in AC092723.1 KD cells (Fig. 4L and Supplementary Fig. 7J), implying that AC092723.1 acts as a scaffold to recruit TET1 to the *IRF8* promoter region.

TET1 is an important chromatin-modifying enzyme that causes DNA demethylation and thus activates gene expression[53]. To test whether TET1 controls *IRF8* expression in this manner, we examined the methylation levels in the *IRF8* promoter region after silencing the expression of AC092723.1. As expected, the methylation level of the promoter region increased significantly

from 6.55 to 14.88% after AC092723.1 KD (Fig. 4M). In conclusion, these data suggest that AC092723.1 interacts with TET1 to limit the methylation levels of the *IRF8* promoter region, leading to the activation of *IRF8* transcription.

**The rs2280381 allele differentially regulates the expression of AC092723.1 and *IRF8* through regulating PU.1 binding and the chromatin status.** The eQTL data suggested an association between the rs2280381 allele and the expression levels of *IRF8* and AC092723.1 (Supplementary Fig. 8A, B), and this association was cell-type-specific in different immune cell subpopulations (Supplementary Fig. 8C). Still, there was no direct evidence that the rs2280381 allele differentially regulates *IRF8* or AC092723.1 expression. To investigate whether the rs2280381 allele directly regulates the expression of AC092723.1 and *IRF8*, we adopted the prime editing technology[54] to generate isogenic cell lines carrying rs2280381 homozygous major (T/T), homozygous minor (C/C), or heterozygous (T/C) alleles (Fig. 5A and Supplementary Fig. 8D). For each genotype, we selected six clones to measure the effect of rs2280381 on *IRF8* and AC092723.1 expression. In agreement with the eQTL data, RT-qPCR data showed that the SLE risk allele T reduced the expression of AC092723.1 and *IRF8* compared with the non-risk allele C (Fig. 5B, C), consistent with the down-regulated expression of *IRF8* and AC092723.1 in SLE patients (Supplementary Fig. 8E–I).

Having demonstrated the allele-specific regulatory ability of rs2280381, we next sought to explore the mechanisms behind it. Genetic variation is often associated with different enhancer activities, which is thought to be an important mechanism for allele-specific regulation of gene expression by SNP alleles. To test whether different rs2280381 alleles can alter the chromatin state, we first analyzed the allelic distribution of H3K27ac at the rs2280381 site through the MARIO pipeline[55] using public ChIP-seq data. We found that rs2280381 showed a strong bias in the direction of non-risk allele for H3K27ac signaling (Fig. 5D), and H3K27ac allele-specific ChIP-qPCR further confirmed these results in rs2280381 heterozygous cell line (Supplementary Fig. 8J). We also examined the chromatin accessibility of the rs2280381 alleles by FAIRE allele-specific qPCR. Consistent with the histone modified allele bias, enhancers carrying the C allele exhibited more FAIRE signal than those carrying the T allele (Fig. 5E), indicating that the C allele has higher chromatin accessibility than the T allele.

Differential TF binding is another feature of the effect of SNP-dependent *cis*-regulation on gene expression. To determine the differential binding of TFs to the rs2280381 sequence, we performed DNA affinity precipitation assay (DAPA) followed by mass spectrometry (MS) experiments, which identified more than 100 candidate proteins. Most of the proteins identified by DAPA-MS were histones or chromatin structure maintenance proteins (Supplementary Data 9). To identify proteins binding to

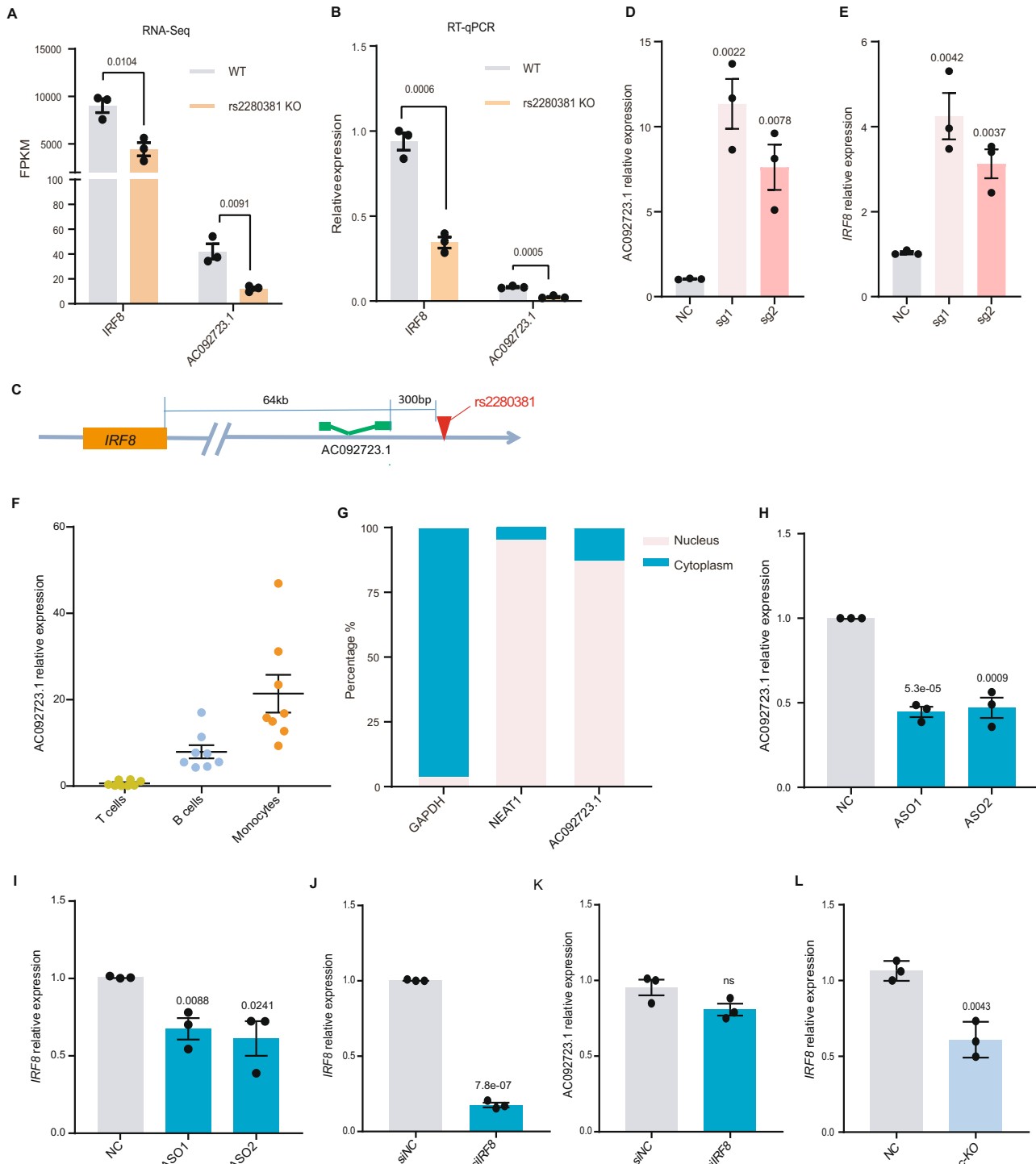

**Fig. 3 AC092723.1 mediates the effect of the rs2280381-containing region on *IRF8* expression. A, B** RNA-sequencing and RT-qPCR analysis of the expression of *IRF8* and AC092723.1 in U-937 WT clones and KO clones (*n* = 3, biologically independent samples). WT: rs2280381 wildtype, KO: 138 bp fragment containing the rs2280381 deletion. **C** The relative location of rs2280381, AC092723.1, and *IRF8*. **D, E** CRISPR SAM assay increases AC092723.1 and *IRF8* expression by targeting the rs2280381-containing region using specific gRNA in U-937 cells (*n* = 3, biologically independent experiments). **F** AC092723.1 expression in different human immune cell subpopulations as measured by RT-qPCR (*n* = 8, biologically independent samples). **G** RT-qPCR analysis of AC092723.1 abundance in nuclear and cytoplasmic fractions of primary monocytes. *GAPDH*, cytoplasmic marker. NEAT1, nuclear marker. **H, I** RT-qPCR analysis of AC092723.1 and *IRF8* expression with or without AC092723.1 knockdown in primary monocytes (*n* = 3, biologically independent experiments). **J, K** RT-qPCR analysis of AC092723.1 and *IRF8* expression with or without *IRF8* knockdown in primary monocytes (*n* = 3, biologically independent experiments). **L** RT-qPCR analysis of *IRF8* expression after deletion of a partial region of AC092723.1 by CRISPR-Cas9 (*n* = 3, biologically independent experiments). Data are represented as mean ± SEM, and *p* values are calculated using an unpaired two-tailed Student's *t* test. ns not significant. See also Supplementary Fig. 6.

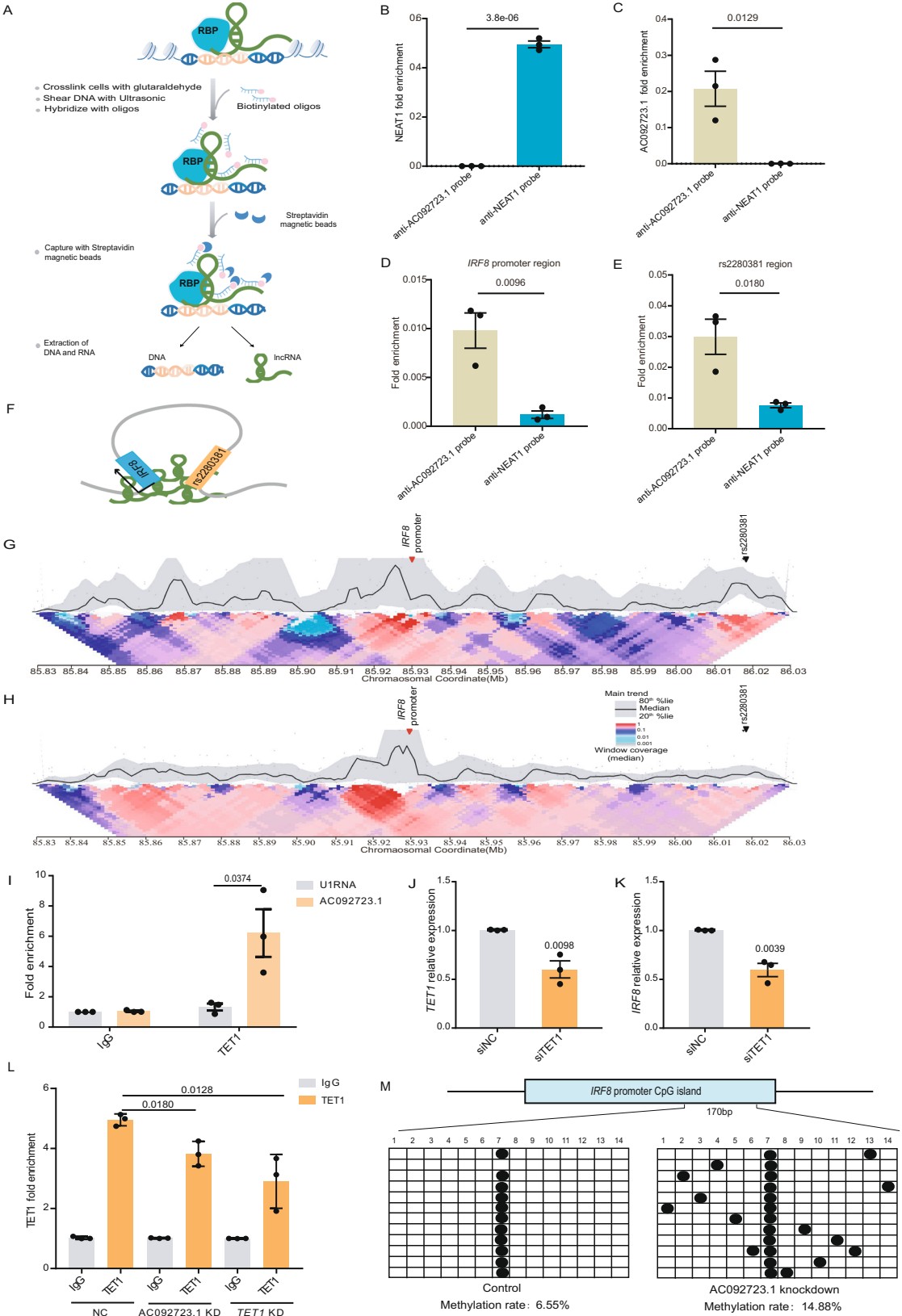

rs2280381, we first compared the proteins pulled down by the rs2280381 sequence to those pulled down using a random control sequence, we focused on the proteins detected only using the rs2280381 for the pull down. In addition, we used ChIP-seq data of Cistrome database[44] to collect all proteins that bind to this site (Supplementary Data 9), and we selected the overlapping proteins

between DAPA-MS and Cistrome database as our candidates. Finally, we focused on monocyte-specific TFs and identified PU.1, an important monocyte lineage-determining TF[56], as our top candidate. ChIP-seq data from primary monocytes showed strong enrichment of PU.1 at this SNP site (Fig. 5F), and our ChIP-qPCR result further verified this binding (Supplementary Fig. 8K).

**Fig. 4 AC092723.1 binds to the *IRF8* promoter to recruit TET1 and affect promoter methylation levels. A** Flow scheme for the ChIRP assay detecting the interaction between the *IRF8* promoter region and AC092723.1, RBP RNA-binding protein. **B, C** NEAT1 RNA and AC092723.1 RNA are enriched explicitly with anti-NEAT1 probes and anti-AC092723.1 probes in ChIRP assay in primary monocytes, respectively ($n = 3$, biologically independent experiments). AC092723.1 interacts with the *IRF8* promoter region (**D**) and rs2280381-containing region (**E**) in primary monocytes ($n = 3$, biologically independent experiments). **F** Model for AC092723.1 contribution to loop formation between the *IRF8* promoter region and the rs2280381 region. **G, H** Knockdown AC092723.1 decreased the interaction frequency between the *IRF8* promoter and the rs2280381 site. 4C-seq analysis of contact profiles between the *IRF8* promoter and the rs2280381 site in primary monocytes without (**G**) or with AC092723.1 knockdown (**H**) using a 2 kb window size in the main trend subpanel. Red arrowhead indicates the view point position, and black arrowhead indicates the target position. Gray dots indicate normalized contact intensities. The heat map displays a set of medians of normalized contact intensities calculated at different window sizes. **I** RIP-qPCR analysis of the interaction between AC092723.1 and TET1 in primary monocytes ($n = 3$, biologically independent experiments). RT-qPCR analysis of *TET1* (**J**) and *IRF8* (**K**) expression in primary monocytes after knockdown of *TET1* by siRNA ($n = 3$, biologically independent experiments). **L** ChIP-qPCR analysis of the binding efficiency of TET1 to the *IRF8* promoter with or without AC092723.1 knockdown in primary monocytes ($n = 3$, biologically independent experiments). KD knockdown. **M** Methylation levels of the *IRF8* promoter region in primary monocytes with or without AC092723.1 knockdown. Data are represented as mean ± SEM, and *p* values are calculated using an unpaired two-tailed Student's *t* test. ns not significant. See also Supplementary Fig. 7.

Moreover, siRNA knockdown of PU.1 expression strongly reduced the expression of AC092723.1 and *IRF8* in primary monocytes (Fig. 5G–I) and U-937 cells (Supplementary Fig. 8L–N). In addition, we detected PU.1 binding to the rs2280381 allele, and AS-ChIP-qPCR data indicated that the C allele had stronger PU.1 binding affinity than the T allele (Supplementary Fig. 8O), which is consistent with the results of luciferase reporter assay (Supplementary Fig. 8P, Q) and the results provided by ADASTRA database[57] (Fig. 5J), reflecting the allele-specific TF binding in human genome. In summary, enhancer with rs2280381 C allele showed stronger signals for PU.1 and H3K27ac, exhibited stronger chromatin accessibility relative to the T allele, and enhanced the expression of AC092723.1 and *IRF8*.

## Discussion

TFs play a critical role in developing autoimmune disease, and disease-associated genetic variants help reveal critical mechanisms involved in disease pathogenesis[12,58]. However, only a few functional genetic variants have been identified to alter TF gene expression levels. To fill this gap, we devised a general strategy to define functional variants that regulate *IRF8* expression-application of this strategy identified rs2280381 as a causal variant at the *IRF8* locus. We also elucidate the specific biological mechanisms by which rs2280381 mediates SLE risk: altered PU.1 binding, H3K27ac enrichment, chromatin accessibility, and lncRNA expression, resulting in distinct *IRF8* promoter methylation levels and altered *IRF8* expression (Fig. 6).

Progress in identifying functional genetic variants that regulate TFs has been continuously challenged by the presence of LD, the specific cell types in which the variants function, and the complexity inherent in deciphering gene regulatory mechanisms[14]. By integrating genetic data and epigenomic analysis, we devised an approach to first rank all SNPs associated with autoimmune diseases in the *IRF8* locus based on the presence of active enhancer marker and then identify candidate SNPs with potential regulatory function. Using the dCas9-VP64 CRISPRa assay, we performed a systematic screening of these genetic variants based on their ability to regulate *IRF8* expression, effectively identifying functional regulatory elements harboring disease-associated SNPs, some of which were further confirmed by SAM CRISPRa and KRAB-dCas9 interference. In general, enhancers can be mapped by CRISPR-mediated deletion, activation, and interference[42,59,60], with KRAB-dCas9-mediated interference of enhancer function arguably the most widely used approach. Notably, the KRAB-dCas9 system has been shown to induce heterochromatin establishment across about 1 kb window around a gRNA's target site[42,61–63]. However, in our study, the distance between several SNPs is less than 500 bp. Therefore, to fine map

the function of regulatory elements carrying disease-associated variants, we chose dCas9-VP64 CRISPRa assay as our preference for screening. We propose that our strategy provides a blueprint for identifying functional SNPs that regulate the expression of genes encoding TFs or other key immune response genes.

Enhancers have been suggested as effective therapeutic targets for disease interventions because targeting enhancers may facilitate precise treatment due to the cell-type specificity of enhancers[11,64–66]. For example, erythroid-specific enhancer of *BCL11A* by CRISPR-Cas9 editing restores γ-globin synthesis for treating sickle cell disease[11]. Thus, the discovery of disease-critical enhancers would provide a valuable therapeutic target for disease treatment. In the present study, using CRISPR-Cas9-mediated deletion, we edited the rs2280381-containing region in different cell lines and different immune cell subpopulations. We found that the rs2280381-containing region acts as a distal and cell-type-specific enhancer to regulate *IRF8* expression, suggesting that the rs2280381 enhancer has the potential to be a future SLE therapeutic target. In this way, deciphering the functional gene variants associated with autoimmune disease will help develop new therapeutic approaches.

Gene expression is controlled by regulatory elements, including distal enhancers and the proximal promoter[67,68]. Distal enhancers interact spatially with promoter regions to regulate the expression of target genes[69]. We performed a 4C-seq assay that validated a promoter-enhancer loop between the *IRF8* promoter and the rs2280381-containing region, which further supports the regulatory function of the rs2280381-containing region. Interestingly, we also observed associations between the *IRF8* promoter locus and various other genomic regions (Supplementary Data 10), some of which contain autoimmune disease-associated genetic variants and the regulatory functions have been validated in CRISPRa screening assays. However, the functions of most regions that exist interactions with *IRF8* promoter remain unknown. Dissecting the role of these regulatory elements will likely help us understand the complete picture of *IRF8* transcriptional regulation.

A major challenge inherent in studying non-coding genetic variants is to validate the functional consequences of different alleles on gene expression. This study generated cell clones carrying two rs2280381 alleles by Prime editing and demonstrated allele-specific regulation of *IRF8* expression by rs2280381. Genetic analyses identified the rs2280381 T allele as a risk allele for SLE[24,27]. Our results showed that rs2280381 risk allele T resulted in lower expression of *IRF8* in monocytes than the non-risk allele C, which is consistent with the eQTL data of primary monocytes in the ImmuNexUT database[70], implying that low expression of *IRF8* in monocytes is a risk factor for SLE. However, the correlation between *IRF8* expression and autoimmunity

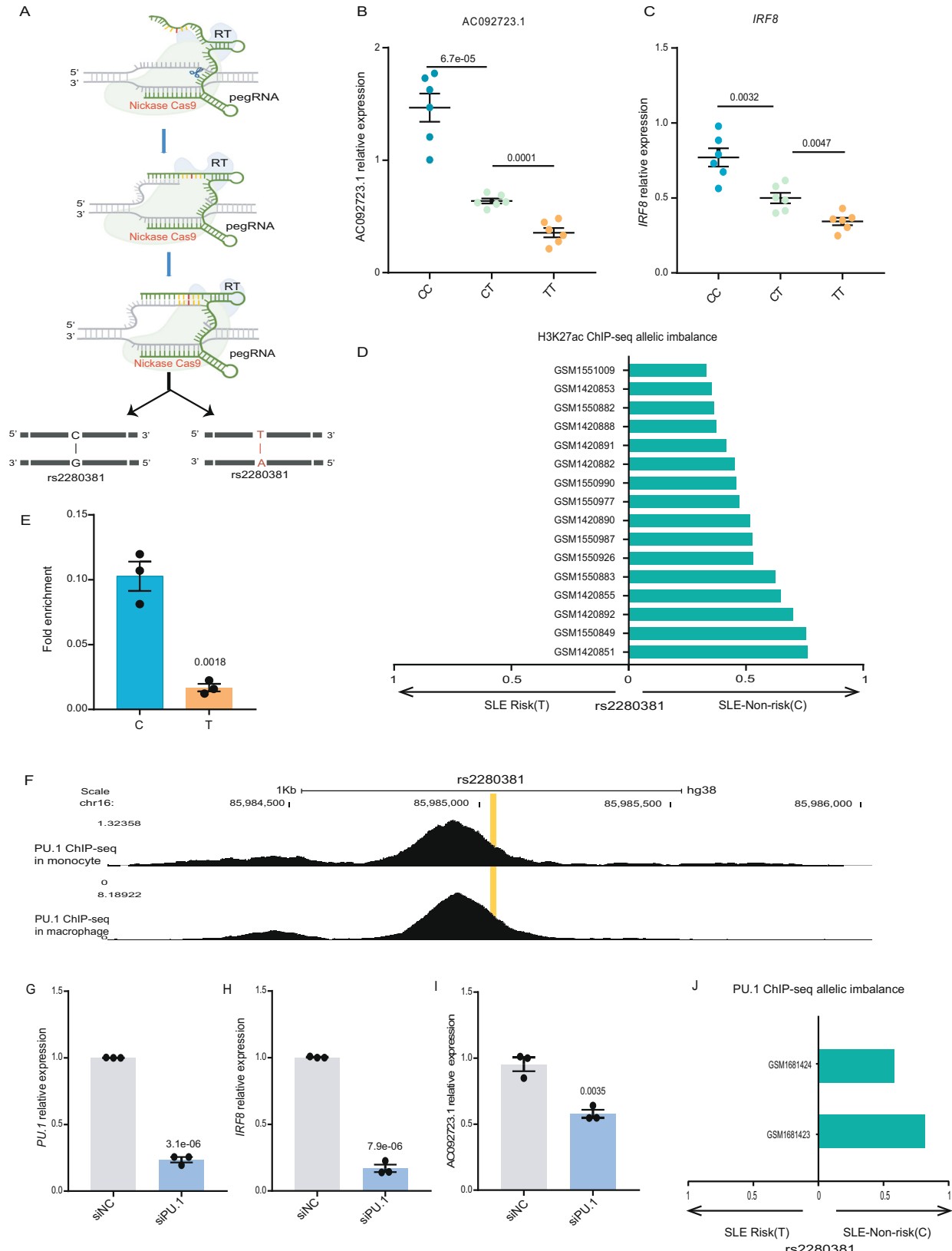

remains controversial, and the available findings are contradictory. For example, *IRF8* deficiency affects plasmacytoid dendritic cell development and reduces the autoimmunity in lupus-prone NZB mice[71]. However, conditional KO *IRF8* in B cells leads to disruption of B cell tolerance, thereby promoting the development of autoimmune disease[72]. Consistent with this

finding, our data, and publicly available RNA analysis data showed decreased expression of *IRF8* in SLE patients (Supplementary Fig. 8F–I). Furthermore, *IRF8* has a dose-dependent effect on the differentiation of DCs, monocytes and neutrophils. Low dose of *IRF8* induces differentiation of pDCs and patrolling monocytes[73], which are the cell types that promote lupus

**Fig. 5 rs2280381 alleles affect H3K27ac and PU.1 binding to fine-tune the expression of AC092723.1 and *IRF8*. A** Workflow for the generation of isogenic cell clones with the Prime editing technology. RT Reverse Transcriptase. **B**, **C** The rs2280381 C allele leads to higher expression of AC092723.1 and *IRF8* compared to the T allele. (*n* = 6, biologically independent samples). **D** Genotype-dependent binding of H3K27ac for rs2280381. Results with MARIO ARS value >0.4 across ChIP-seq datasets are included. The *X*-axis indicates the preferred allele, along with a value indicating the strength of the allelic behavior, calculated as one minus the ratio of the weak to strong read counts (e.g., 0.5 indicates the strong allele has twice the reads of the weak allele). ARS allelic reproducibility score. **E** The genomic region containing the non-risk allele C exhibits increased chromatin accessibility compared to the risk allele T, as determined by AS-FAIRE-qPCR in the rs2280381 heterozygous U-937 cell clone. (*n* = 3, biologically independent experiments). **F** ChIP-seq in monocyte and macrophage indicates the PU.1 binds to rs2280381 site. Relative expression of *PU.1* (**G**), *IRF8* (**H**) and AC092723.1 (**I**) after PU.1 siRNA-mediated knockdown, as measured by RT-qPCR in primary monocytes (*n* = 3, biologically independent experiments). **J** PU.1 prefers binding to the rs2280381 C non-risk allele, as analyzed by MARIO methods using the PU.1 ChIP-seq data in primary monocytes or macrophage provided by ADASTRA database. Data are represented as mean ± SEM, and *p* values are calculated using an unpaired two-tailed Student's *t* test. ns not significant. See also Supplementary Fig. 8.

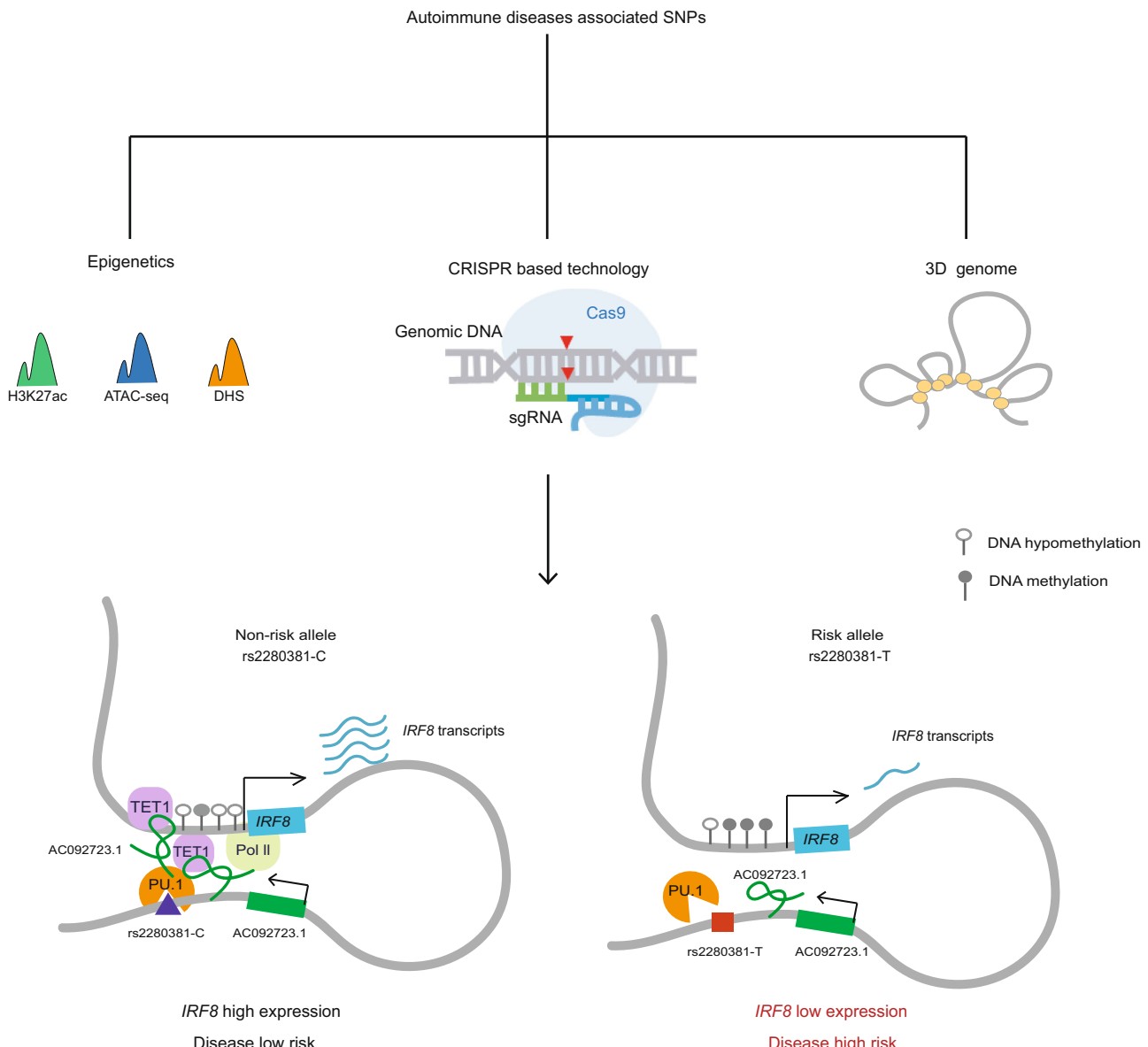

**Fig. 6 Model for rs2280381 regulates *IRF8* expression mediating disease risk.** The rs2280381-containing region forms a gene-loop with the *IRF8* promoter region. The rs2280381 T risk allele has lower PU.1-binding affinity than the non-risk C allele, which results in the reduced expression of AC092723.1, upregulation of methylation levels at *IRF8* promoter, and the decreased expression of *IRF8* contributing to SLE risk. DHS DNase I hypersensitive site.

pathogenesis[71,74]. In conclusion, the role of *IRF8* in SLE is highly complicated. The function of *IRF8* in SLE may be stage- or cell-type-specific, which still needs to be carefully deciphered in the future.

Most individual SNPs have only a tiny effect on gene expression or disease-associated phenotypes. Several studies have found that genetic variants within multiple enhancers of a gene can synergistically regulate gene expression, thereby amplifying these individual minor effects[75]. In our study, in addition to the region containing rs2280381, we found several other regions containing genetic variants also increase *IRF8* expression in CRISPRa screening assays. These data suggest that combinations of functionally independent genetic variants may be an important risk factor for disease. To fully unravel the mechanisms of genetic-mediated disease risk, the synergistic effects of multiple genetic variants should be emphasized in future studies.

Allele-dependent TF binding is a major factor contributing to allelic expression differences. Using DAPA-MS data, ChIP-qPCR and ChIP-Seq data, we identified PU.1 as a critical TF that binds to the rs2280381 locus. There are differences in the binding of PU.1 to the rs2280381 non-risk allele and risk allele, which may lead to different regulatory functions of the risk and non-risk allele. PU.1 is the key TF that elicits monocyte-specific enhancer of a key linear-determining transcription factor (LDTF)[56] and the binding of PU.1 to the rs2280381 locus may contribute to the establishment of cell-type-specific enhancer at this locus. In addition, we observed different chromatin states for the risk and non-risk alleles, as reflected by the higher H3K27ac enrichment and chromatin accessibility of the non-risk C allele compared to the T risk allele. Collectively, these observations shed light on the possible mechanisms of rs2280381 risk allele-mediated disease risk. Notably, our DAPA experiments also revealed binding of several other proteins to rs2280381 (Supplementary Data 9), including LDTFs and chromatin regulators. Whether these factors contribute to the allele-specific regulation of rs2280381 on *IRF8* expression, and if so, which mechanism they use to direct the cell-type-specific enhancer activation at this site still will be the subject of future studies.

Intriguingly, our results suggest that the eRNA AC092723.1 is involved in the regulatory mechanism of the differential effect of the rs2280381 allele on *IRF8* expression. We show that the rs2280381 allele is associated with differential expression of AC092723.1, which can directly regulate the expression level of *IRF8*. LncRNA can regulate chromatin structure, participate in remodeling complexes, and modify chromatin to enhance or repress gene expression[49,50,76,77]. Our ChIRP assay shows that AC092723.1 can bind to both the rs2280381-containing region and the *IRF8* promoter site. That knockdown of AC092723.1 reduces the connected frequency between the rs2280381-containing region and the *IRF8* promoter, suggesting that AC092723.1 promotes loop formation, which provides direct evidence that lncRNA regulates chromatin structure. Using a combination of bioinformatics-based predictions and RIP-qPCR assays, we identified TET1 as a binding partner of AC092723.1. TET1 is a key chromatin modifier that regulates gene expression by affecting DNA methylation levels[53]. Our ChIRP and ChIP assays indicate that recruitment of TET1 by AC092723.1 leads to TET1 binding to the *IRF8* promoter and reducing DNA methylation levels, thereby regulating *IRF8* expression. Changes in DNA methylation contribute to the pathogenesis of SLE, but the factors that regulate SLE methylation are mainly unknown. Our study links an SLE-associated genetic variant to DNA methylation and ultimately to SLE etiology, adding another layer of regulation to the genetic variant-based modulation of gene expression involved in the disease.

In conclusion, our study provides a blueprint for establishing a link between disease risk genetic variants and TF gene expression levels and applying this approach to decipher an essential mechanism of SLE risk SNP-mediated disease pathogenesis. Our work also provides critical insights that lay a solid foundation for developing enhancer-based disease therapies.

## Methods

**Cell culture**. All cell lines were purchased from the Chinese academy of science cell bank (Shanghai, China). U-937 (TCHu159), Raji (TCHu141) and Jurkat (TCHu123) were cultured in 10% (v/v) fetal bovine serum (FBS) and 90% RPMI-1640 medium. HEK-293T (GNHu17) was cultured with 10% (v/v) FBS and 90% Dulbecco's Modified Eagle Medium. Cells were maintained at 37 °C and 5% $CO_2$ constant temperature incubator. These cell lines were free of mycoplasma during our study.

**RNA extraction and RT-qPCR**. Total RNA was extracted using TRIzol Reagent (Invitrogen). Samples of 500 ng of RNA were reverse transcribed using Prime-Script™ RT Reagent Kit (Perfect Real Time) (TAKARA, RR037A). qPCR was performed using TB Green Premix Ex Taq reagent (TAKARA, RR420A) and data were collected by QuantStudio Real-Time PCR software (ABI, v1.2). GAPDH expression was determined as an internal control and fold-change in expression level was calculated using the $\triangle\triangle$Ct method.

**Western blotting**. Protein lysates were separated on 10% SDS/PAGE gels, transferred to PVDF membranes and probed with antibodies directed against IRF8 (Cell Signaling Technology, 5628S, 1:1000 dilution), HRP-conjugated GAPDH (Abcam, AC035, 1:5000 dilution). GAPDH was used as a loading control.

**ASOs and siRNAs transfection**. ASOs and siRNAs were synthesized by Sangon Biotech (Shanghai, China). Before transfection, $2 \times 10^5$ cells were seeded into a 24-well plate and incubated at 37 °C and 5% $CO_2$ for overnight. Next, 200 nM of ASO or siRNA were transfected into the cells using TransIntroTM EL Transfection Reagent (Transgene, FT201-01) and cells were collected to extract RNA. For primary cells, ASO or siRNA were transfected with NEON system.

**RNA immunoprecipitation (RIP)-qPCR**. RIP assays were performed using the EZ-Magna RIP Kit (Millipore). Briefly, $1 \times 10^7$ cells were lysed with RIP lysis buffer. Cell extracts were co-immunoprecipitated with Anti-TET1 antibody (Active motif, 61741, 2 μg per $10^7$ cell lysate), Anti-KAT3B/p300 antibody (Abcam, ab10485, 2 μg per $10^7$ cell lysate), Anti-WDR5 antibody (Abcam, ab56919, 2 μg per $10^7$ cell lysate), and Anti-DOT1L Antibody (Thermo Fisher, A300-953A, 2 μg per $10^7$ cell lysate). The recovered RNA was subjected to RT-qPCR analysis and U1 was used as a nonspecific control target.

**Candidate SNP picking**. LD expansion was done by HaploReg v4.1 tool (https://pubs.broadinstitute.org/mammals/haploreg/haploreg.php) to include all SNPs in strong LD ($r^2 > 0.8$) with the reported tag SNPs at the *IRF8* locus. The chromatin landscape of SNP-located regions was analyzed using the public resource provided by the NIH Roadmap Epigenomics Mapping Consortium (http://www.roadmapepigenomics.org/). And SNP-located regions with any signal of ATAC-seq peaks, H3K27Ac peaks or DNase peaks in four major human immune cell subsets were selected as candidate SNPs to undergo CRISPRa screening assays.

**Cell fractionation**. This assay was performed using the Nuclear and Cytoplasmic Extraction Kit (CWBIO, Shanghai). In brief, $1 \times 10^7$ cells were harvested and resuspended in 1 ml of Nc-buffer A supplement with 55 μl of Nc-buffer B and incubated on ice for 10 min. Cells were then centrifuged for 15 min at $12,000 \times g$ and the supernatant was collected as the cytoplasmic fraction. The remaining pellets were resuspended in 500 μl of Nc-buffer C supplemented with RNase inhibitors and incubated for 40 min on ice, then sample was centrifuged at $12,000 \times g$ for 15 min and the supernatant was collected as the nuclear fraction. All fractions were resuspended in TRIzol to extract RNA.

**Chromatin immunoprecipitation (ChIP)-qPCR**. This assay was carried out using the SimpleChIP® Plus Enzymatic Chromatin IP Kit (Cell Signaling Technology). Briefly, $5 \times 10^6$ cells were first cross-linked by 1% formaldehyde solution and then quenched by 125 mM glycine solution. After that, cells were washed by cold PBS for twice, resuspended with 1 ml cold $1 \times$ Buffer A, incubated on ice for 10 min and centrifuged to remove supernatant. Then the pellets were resuspended in 1 ml cold $1 \times$ Buffer B, centrifuged to remove supernatant and resuspended in 100 μl $1 \times$ Buffer B. In total, 0.5 μl of Micrococcal Nuclease was added and the sample was incubated at 37 °C for 20 min to digest DNA into 150–900 bp length. After that, 10 μl of 0.5 M EDTA was added to stop the digestion, and samples were centrifuged to get the cell pellets. Finally, pellets were resuspended with 100 μl of $1 \times$ ChIP Buffer and subsequently sonicated at 4 °C with a Bioruptor sonicator (Diagenode) at high power for five cycles with 30 s ON and 30 s OFF. The sonicated samples were centrifuged to discard the debris. The supernatant was collected and

incubated with anti-H3K27ac antibody (ab177178, Abcam, 2 μg for 25 μg of chromatin) or anti-PU.1 antibody (2266S, Cell Signaling Technology, 1:50) or anti-TET1 antibody (61741, Active motif, 4 μl per ChIP assay) for overnight at 4 °C on rotation. ChIP-grade protein A + G magnetic beads (Millipore, 16-663) were added and the enriched chromatin was eluted with 150 μl ChIP Elution Buffer. DNA fragments were purified with spin columns and enrichment was detected by qPCR.

**Formaldehyde-assisted isolation of regulatory elements (FAIRE)-qPCR**. To perform FAIRE-qPCR assay, $1 \times 10^7$ cells were cross-linked by 1% formaldehyde solution and quenched by 125 mM glycine solution. Cells were then sonicated, equal volume of phenol/chloroform/isoamyl alcohol was added into the chromatin lysate and centrifuged to isolate the aqueous. The aqueous was further purified by adding chloroform/isoamyl alcohol. Then DNA was precipitated, washed and reverse cross-linked to prepare the FAIRE DNA. FAIRE DNA samples were analyzed by quantitative RT-PCR with specific primers targeting DNA sequences at different distances to rs2280381 site. Values were normalized to input DNA and compared to a region just outside the putative regulatory region.

**Allele-specific qPCR**. We designed AS-qPCR primers to specifically amplify the rs2280381 region with a T or C allele in the ChIP or FAIRE DNA samples. AS-qPCR was performed according to normal qPCR procedures.

**RNA library preparation, sequencing and gene expression analysis**. Total RNA was extracted using TRIzol Reagent. rRNA was depleted from total RNA using Ribo-Zero™ rRNA removal Kit and library was made using Illumina NEBNext® Ultra™ Directional RNA Library Prep Kit (E7420L, NEB). The libraries were loaded on an Illumina HiSeq X ten instrument (Illumina). Sequencing was carried out using a $2 \times 150$ paired-end configuration, image analysis and base calling were carried out by the HiSeq Control Software (v3.3.76.1) + OLB (v1.9.3) + GAPipeline-1.6 (Illumina) on the HiSeq instrument. Computational analysis of paired-end reads was conducted using cutadapt (v1.15), Samtools (v0.1.19), Hisat2 (v2.1.0), and HT-seq (v0.11.2) software. Statistical normalization and differential analysis were performed in R using the DESeq2 (v1.24.0) package. The threshold to define up or down regulation was log2 fold-change ≥1 and FDR < 0.05. Visualization was also conducted in R (v3.3.3).

**Chromatin isolation by RNA purification (ChIRP)-qPCR[48]**. Probes used in this assay were designed using an online tool (singlemoleculefish.com). Oligonucleotides were synthesized and the 3′ end of the oligonucleotides was biotinylated. To prepare the ChIRP samples, $2 \times 10^7$ cells were first cross-linked in 1% glutaraldehyde solution at room temperature, then quenched with 1.25 M glycine. Cell pellets were washed by cold PBS, resuspended in ChIRP lysis buffer and sonicated into 100–500 bp length using Bioruptor sonicator (Diagenode). After that, 20 μl of lysate was removed to prepare input RNA and DNA sample, then 2 ml hybridization buffer and total 100 pmol probes were added to the remaining lysate, after incubation at 37 °C with gentle shaking for 4 h, 100 μl Dynabeads™ MyOne™ Streptavidin C1(Invitrogen, 65001) were added, and incubated at 37 °C with gentle shaking for 0.5 h to isolate the chromatin. Finally, chromatin samples for isolating RNA were treated with proteinase K and RNA was extracted using TRIzol reagent, the RNA was reverse transcribed into cDNA, the enrichment of AC092723.1 and NEAT1 was detected by RT-qPCR. For isolating DNA, chromatin samples were treated with RNase A and proteinase K, and purified with phenol/chloroform/isoamyl. DNA samples were directly utilized as a template to detect the enriched region.

**Genome editing in cell lines**. For Prime editing, pegRNA was designed using the online CRISPR tool (http://pegfinder.sidichenlab.org/). For constructing nicking gRNA expression vector, pKLV-U6gRNA(BbsI)-PGKpuro2ABFP(Addgene, 50946) was linearized by BbsI (NEB, R3059L) and then gel purified. Guide RNA oligos were synthesized in Tsingke (Shanghai, China), annealed and subcloned into the linearized 50946 plasmid and were transformed into chemically competent Escherichia coli (Stbl3, Transgen Biotech) to extract plasmid DNA. For constructing pegRNA expression vector, 50946 plasmid was cut by BbsI (NEB, R3059L) and BamHI (NEB, R0136S) and then gel purified. Guide RNA oligos, gRNA scaffold oligos, RT temple and prime binding sequence oligos were annealed and subcloned into the BbsI and BamHI cut plasmid. For editing, $2 \times 10^6$ U-937 cells were prepared and washed by PBS for electro-transfection, 10 μg pCMV-PE2-P2A-GFP (Addgene, 132776) plasmid, 10 μg plasmid expressing pegRNA and 5 μg plasmid expressing nicking gRNA were added to cells and resuspend with 100 μl buffer R, then cells were transfected with the condition of 1400 v, 10 ms, three pulses using Neon system. Cells were immediately plated to 6-well plate and cultured for 72 h. Single cell with strong GFP and BFP signals were sorted into the 96-well plate containing 200 μl culture medium in each well by FACS. After 14 days culture, clones were transferred to a 24-well plate and genotype was identified by sanger sequencing.

To delete the genome sequence containing rs2280381, we utilized a dual-guide RNA strategy using two Cas9-guide RNA constructs. gRNAs were designed using CHOPCHOP (version 3) (https://chopchop.cbu.uib.no/#), 1 pair of gRNAs around

rs2280381 with the highest editing efficiency and a relatively lower off-target rate was chosen. gRNA oligos were annealed and subcloned into the BbsI linearized px458 vector (Addgene,48138). In total, $2 \times 10^6$ cells were transfected with 5 μg px458-gRNA1 and 5 μg px458-gRNA2 plasmids using Neon system. Single cell with strong GFP signal were sorted into a 96-well plate by FACS. After 14 days of culture, genomic deletions were screened by Sanger sequencing of PCR amplicons. Electroporation conditions for each cell line were as follows: U-937, 1400 v, 10 ms, 3 pulses; Raji, 1350 v, 30 ms, 1 pulse; Jurkat, 1350 v, 10 ms, 3 pulses.

**CRISPRa screening**. To design CRISPRa gRNAs, we first downloaded candidate SNP-centered 200 bp length sequences from human genome build GRCh38/hg19 (https://genome.ucsc.edu/cgi-bin/hgGateway) and utilized the CHOPCHOP online gRNA design tool (https://chopchop.cbu.uib.no/#) to obtain gRNA according to higher efficiency and lower off-target rate. For each candidate SNP, 3 gRNAs were designed around the SNPs and synthesized by GenScript Inc. The gRNA was dissolved to 35 μM concentration and stored at −20 °C. Prior to delivering gRNA into cells, U-937 cells stably expressing dCas9-VP64-Blast (Addgene, 61425) and MS2-P65-HSF1-Hygro (Addgene, 61426) were established by transduction of corresponding lentivirus following selection with 10 μg/ml Blasticidin (Invivogene, ant-bl-5) and 300 μg/ml Hygromcin (Thermo Fisher, 10687010) for 1 week. For screening, $2 \times 10^5$ U-937 cells were resuspended in Buffer R, and 0.5 μl of each gRNA targeting the corresponding SNP were added into the cells. Then, the gRNA-Cells-buffer R mixture was aspirated into the 10 μl Neon pipette tip, and transfected using the Neon transfection system with the condition 1400 V, 10 ms, 3 pulses. After transfection, the cells were immediately transferred into a 24-well plate containing pre-warmed 10% FBS + 90% RPMI-1640 media. After 24 h culture, cells were collected to extract RNA.

**CRISPR SAM assay in the U-937 cell line**. gRNAs targeting the rs2280381-containing region were synthesized, annealed and cloned into lenti-sgRNA(MS2)-zeo backbone plasmid (Addgene, 61427) using restriction enzyme BsmBI (NEB, R0580L). gRNA lentivirus particles were produced and transduced into a U-937 cell line stably expressing dCas9-VP64 and MS2-P65-HSF1 fusion proteins. After transduction, cells were selected with 400 μg/ml Zeocin (R25001, Thermo Fisher) for 72 h and RNA was collected to detect gene expression level.

**KRAB-dCas9 assay in the U-937 cell line**. This assay was performed similarly to the CRISPR SAM activation assay. Briefly, U-937 cell line stably expressing KRAB-dCas9-mCherry was generated using the pHR-SFFV-KRAB-dCas9-P2A-mCherry plasmid (Addgene, 60954). Then gRNA lentivirus particles targeting the rs2280381-containing region were produced and transduced into the cells. After transduction, cells were selected with 400 μg/ml Zeocin (R25001, Thermo Fisher) for 72 h and RNA was collected to detect gene expression level.

**PBMC isolation**. Healthy human donors and SLE patients were recruited and signed informed consent according to the internal review and ethics boards of Renji Hospital, Shanghai Jiao Tong University (SJTUSM). The experiments were approved by the internal review and ethics boards of Renji Hospital, Shanghai Jiao Tong University School of Medicine (SJTUSM). PBMCs were isolated using Ficoll-Paque density gradient solution (density = 1.077 g/ml; GE Healthcare). Peripheral blood was mixed in a 1:2 ratio with phosphate-buffered saline (PBS) containing 2% FBS and 2 mM EDTA. After density gradient centrifugation ($400 \times g$, 35 min, no brakes), the PBMC layer was carefully removed and the cell pellets were washed twice with PBS for further study.

**Lentivirus production**. In total, $3 \times 10^5$ HEK-293T cells were seeded into a six-well plate and incubated at 37 °C and 5% $CO_2$ for overnight. Then cells were transfected with 1 μg of targeting plasmid, 250 ng of pMD2.G (Addgene, 12259), and 750 ng of psPAX2 (Addgene, 12260) using 3 μl of Lipofectamine 2000 (Thermo Fisher, 11668-019). The media was changed after transfection for 6 h. After transfection for 72 h, virus supernatant was collected and centrifuged at 4 °C for 10 min to remove the debris. The supernatant was aliquoted, and stored at −80 °C.

**MARIO analysis of allelic ChIP-seq reads**. MARIO (v3.93)[55,78] method was used to analyze the ChIP-seq data. In brief, rs2280381 is heterozygous in the assayed cell line and located within a peak in a given ChIP-seq dataset was identified by MARIO. Then, the sequencing reads mapping to each heterozygote in the peak for imbalance between the two alleles was examined. The Allelic Reproducibility Score (ARS) was produced by combining across experimental replicates, and MARIO ARS value >0.4 and pass the following criteria were considered allelic: (1) rs2280381 must be significantly allelic for in at least 50% of the ChIP-seq datasets in which that rs2280381 was heterozygous; (2) the variant shows significant allelic behavior must be strongly preferred (ARS > 0.4) in at least 75% of the datasets for the same allele; (3) the same strong allele must be preferred in the replicates of a given experiment.

**Circular chromatin conformation capture assay (4C) sequencing**. To perform 4C-seq experiments, $1 \times 10^7$ cells were collected and cross-linked by 1% formaldehyde

solution. Then cells were quenched by 125 mM glycine solution. Cell pellets were resuspended in 5 ml cold lysis buffer (50 mM Tris, 150 mM NaCl, 5 mM EDTA, 0.5% NP-40, 1% Triton X-100, 1× protease inhibitor) and incubated on ice for 10 min. After lysis, cell nuclei pellets were collected, washed and resuspended in 500 μl 1× Csp6I buffer. In total, 15 μl 10% SDS was added and incubated for 1 h in a shaker at 750 r.p.m, then 75 μl 20% Triton X-100 was added and incubated for another 1 h with gentle shaking to sequester the SDS. In total, 200 units of Csp6I enzyme (Thermo Fisher, FD0214) (for rs2280381 view point, Csp6I was replaced with MboI (NEB, R0147M)) were added for a 4 h incubation at 37 °C in a shaker at 900 r.p.m. Then, 200 units of Csp6I enzyme was re-added and incubated at 37 °C in a shaker at 900 r.p.m for overnight. Enzyme was inactivated at 65 °C for 20 min, 700 μl 10×T4 DNA ligase buffer was added and supplemented with Milli-Q ddH$_2$O to a total volume of 7 ml. Then, 100 units of T4 DNA ligase was added and incubated at room temperature for 6 h. After that, 30 μl of Proteinase K (10 mg/ml) was added and incubated at 65 °C for overnight. The remaining RNA was cleared by adding 30 μl RNase A (10 mg/ml) and incubating at 37 °C for 45 min. DNA was extracted with equivalent phenol/chloroform/isoamyl, and the pellets were dissolved in 150 μl 10 mM Tris-HCl (pH 7.5). The chromatin was further digested with 50 units of NlaIII enzyme in 500 μl volume and incubated at 37 °C for overnight. After that, enzyme was inactivated at 65 °C for 20 min. The digested chromatin was ligated for the second time using 100 Units of T4 DNA ligase, 1.4 ml 10 × T4 DNA ligation buffer, supplemented with ddH$_2$O to 14 ml volume, and incubated at room temperature for 4 h. DNA was purified using phenol–chloroform and further purified with the QIAquick PCR purification kit (Qiagen, 28106). The DNA concentration was detected by Qubit (Thermo Fisher). Finally, the 4C-seq library was constructed by amplification of template using the 2×High-Fidelity Master Mix kit (Tsingke, TP001) with locus-specific primers containing Illumina sequences. The libraries were purified and sequenced on a HiSeq × ten (Illumina). 4C-seq data were analyzed using the software pipeline 4Cseqpipe (version 0.7), with settings: -stat_type median, trend_resolution 2000. Normalized trend was computed within the genomic region (chr16: 85,860,001–86,060,000) for both viewpoints. Bowtiealign (version 1.2) was used to map captured reads to the Homo sapiens genome assembly GRCh37 (hg19) with the settings: -m 1 and captured fragments on chromosome 5 (reads per million more than 20) were listed. Circos plot were created with circus (v0.69-6).

**Cas9 RNP assembly.** Alt-R crRNAs and Alt-tracrRNA-ATTO550 (IDT, 1075928) were ordered from Integrated DNA Technologies (IDT) and dissolved with Nuclease-Free Duplex Buffer (IDT) to 200 μM concentration. Equimolar concentrations of two oligos were mixed to a final 44 μM concentration and annealed. For each reaction, 22 pmol of crRNA-tracrRNA duplex and 18 pmol of HiFi Cas9 protein (1081061, IDT) were mixed in Buffer T to a final volume 1 μl and incubated at room temperature for 10 min to prepare the Cas9 RNP.

**Primary immune cell subset isolation and editing.** CD3+ T cells, CD14+ monocytes and CD19+ B cells were isolated from human PBMCs using the Human CD3+ T-Cell Isolation kit (Miltenyi Biotec,130-050-101), Human CD14+ monocytes Isolation kit (Miltenyi Biotec, 130-050-201) and Human CD19+ B Cell Isolation kit (Miltenyi Biotec, 130-050-301) respectively.

For T-cell editing, after isolation, T cells were cultured in OpTmizer™ CTS™ T-Cell Expansion SFM medium (Thermo Fisher, A10458-03) supplemented with CD3/CD28 dynabeads (Thermo Fisher, 11131D) for 48 h. Before transfection, CD3/CD28 dynabeads were removed and T cells were cultured for another 6 h. Then 2 × 10$^5$ cells were washed twice with PBS and resuspended into 9 μl of Buffer T, mixed with Cas9 RNP and electroporated using the Neon transfection system with the condition 1400 V, 10 ms, 3 pulses. After that, T cells were transferred to the culture medium supplemented with 30 IU/ml IL-2 (Peprotech, 200-02 A). After electroporation for 3 days, cells were collected to extract RNA and DNA.

For B cell editing, after isolation, B cells were cultured in RPMI-1640 medium with 10% (vol/vol) HI-FBS, 2 mM L-Glutamine, 55 μM β-mercaptoethanol, 50 IU/ml interleukin 4 (Peprotech, 200-04) and supplemented with CD40 ligand (Miltenyi Biotec, 130-098-775) for 48 h. In total, 1.2 × 10$^5$ cells were collected and washed twice with PBS and resuspended into 9 μl of Buffer T, mixed with Cas9 RNP and electroporated using the Neon transfection system with the condition 1400 V, 10 ms, 3 pulses. After transfection, cells were immediately transferred to 500 μl of culture medium and cultured for 3 days, then cells were sorted to extract RNA and DNA.

For monocyte editing, 2.5 × 10$^5$ monocytes were washed twice with PBS and resuspended into 9 μl of Buffer T, mixed with Cas9 RNP, 1 μl Alt-R Cas9 Electroporation Enhancer and electroporated using the Neon transfection system with the condition 1600 V, 10 ms, 3 pulses. Cells were immediately transferred to 200 μl of medium containing 90% RPMI-1640 medium, 10% (vol/vol) HI-FBS, 2 mM L-Glutamine and 55 μM β-mercaptoethanol. After electroporation for 24 h, cells with strong ATTO550 signal were collected to extract RNA and DNA.

**DNA methylation analysis.** DNA was extracted from cells using the QIAamp DNA Mini Kit (Tiangen, DP304) and quantified using NanoDrop. Then 1 μg genomic DNA was treated with sodium bisulfite by using the EZ DNA Methylation-Gold Kit (Zymo, D5006). Purified bisulfite-converted DNA was amplified using the 2 × High-Fidelity Master Mix (Tsingke, TP001). PCR products were gel purified and cloned into the pGEM®-T Easy Vector vector (Promega,

A1360). Cloning products were transformed into competent E. coli cells and plated on carbenicillin plates with X-gal for blue-white screening. Colonies were picked and sequenced by Sanger sequencing to analyze the *IRF8* promoter methylation level.

**Luciferase reporter assay.** The *IRF8* promoter (934 bp, −724 to +210 relative to the TSS, chr16: 85,898,444-85,899,377, hg38) was selected on the basis of the CpG island, amplified and inserted into the pGL3-basic vector (U47295, Promega) on Hind III site using the Gibson assembly cloning method (10911ES20, Yeasen, Shanghai, China) to construct *IRF8* promoter-pGL3 reporter plasmid. The rs2280381-centered sequence (219 bp, chr16:85,984,914–85,985,132, hg38) was amplified with the genomic DNA carrying the rs2280381 C allele or T allele and inserted into the *IRF8* promoter-pGL3 reporter plasmid with SacI and NheI site. For plasmid transfection, 1 × 10$^4$ HEK-293T cells were seed in the white 96-well tissue culture plates and cultured for overnight. Then cells were transfected with 100 ng *IRF8* promoter-pGL3 or rs2280381-*IRF8* promoter pGL3 luciferase reporter plasmid and 2.5 ng Renila luciferase control reporter plasmid using Lipofectamine 2000 (11668-019, Thermo Fisher). After 72 h, cells were analyzed for luciferase activity using the Dual-Glo Luciferase Assay System (E2940, Promega).

**DNA affinity precipitation assay (DAPA)-mass spectrometry.** In brief, U-937 cells were first lysed to extract the nuclear lysates, 200 μg nuclear extracts were mixed with 50 pmol of 5′-biotinylated DNA probes in the Buffer (20 mM HEPES, pH 7.9, 10% glycerol, 50 mM KCl, 0.2 mM EDTA, 1.5 mM MgCl$_2$,100 μg/ml Sheared Salmon sperm DNA, 1 mM dithiothreitol, and 0.25% Triton X-100) and incubated on ice for 45 min. Then Dynabeads™ M-280 Streptavidin (11205D, Thermo Fisher) was added and rotated for 2 h at 4 °C. Then, the enriched proteins were dissociated by the addition of 2 x Laemmli sample buffer (161-0737, Bio-Rad) and boiled at 95 °C for 10 min. The boiled protein samples were digested by trypsin for MS analysis.

**Liquid chromatography-MS/MS analysis and data processing.** The tryptic peptides were dissolved in 0.1% formic acid (solvent A), and directly loaded onto a home-made reversed-phase analytical column (15 cm length, 75 μm i.d.). The gradient comprised an increase from 6 to 23% solvent B (0.1% formic acid in 98% acetonitrile) over 16 min, 23 to 35% in 8 min and climbing to 80% in 3 min then holding at 80% for the last 3 min, all at a constant flow rate of 400 nl/min on an EASY-nLC 1000 UPLC system.

The peptides were subjected to NSI source followed by tandem MS (MS/MS) in Q Exactive™ Plus (Thermo Fisher) coupled online to the UPLC. The electrospray voltage applied was 2.0 kV. The *m/z* scan range was 350–1800 for full scan and intact peptides were detected in the Orbitrap at a resolution of 70,000. Peptides were then selected for MS/MS using NCE setting as 28 and the fragments were detected in the Orbitrap at a resolution of 17,500. A data-dependent procedure was used that alternated between one MS scan followed by 20 MS/MS scans with 15.0 s dynamic exclusion. Automatic gain control was set at 5E4.

The resulting MS/MS data were processed using Mascot Daemon (version2.3.0). Tandem mass spectra were searched against 2019-uniprot-human database. Trypsin/P was specified as cleavage enzyme allowing up to two missing cleavages. Mass error was set to 10 p.p.m. for precursor ions and 0.02 Da for fragment ions. Carbamidomethyl on Cys were specified as fixed modification and oxidation on Met was specified as variable modification. Peptide confidence was set at high and peptide ion score was set >20. We used the following criteria to choose the candidate binding proteins: (1) proteins must be detected only in experiment group; (2) proteins should be TFs based on our group found that TFs occupy multiple loci associated with complex genetic disorders[55]; (3) proteins should be overlapped with ChIP-seq data in this site provided by Cistrome project[44].

All oligos used in this paper are listed in Supplementary Tables in Supplementary Information.

**Statistical analysis.** All statistical analyses were performed using R Studio (version 1.0.136) with R version 3.3.3 and GraphPad Prism 8 software. Data are shown as mean ± SEM. "*n*" represents the number of biologically independent samples or experiments unless otherwise mentioned. Details of the statistical analysis for each experiment can be found in the relevant figure legends. All statistical analyses were calculated using a paired or unpaired two-tailed Student's t test as indicated in the figure legend unless otherwise mentioned.

**Reporting summary.** Further information on research design is available in the Nature Research Reporting Summary linked to this article.

## Data availability

The RNA-sequencing data generated in this study have been deposited in the ArrayExpress database at EMBL-EBI under accession code E-MTAB-11306. The 4C sequencing data generated in this study have been deposited in the ArrayExpress database at EMBL-EBI under accession code E-MTAB-10120 and E-MTAB-11175. The mass spectrometry proteomics data generated in this study have been deposited in the PRIDE database under accession code PXD029688. The public ATAC sequencing data E-MTAB-8982[66] were used to analyze the

chromatin accessibility of primary immune cells. The data from GSE45291[79], GSE108497[80] and GSE110174[81] were used to analyze *IRF8* expression in health control and SLE patients. The data from GSE149050[82] were used to analyze the expression of AC092723.1 in different immune cell subpopulations. GSE155555[43] data were used to analyze the ATAC-seq and H3K27ac signal of U-937 cells. The NIH Roadmap Epigenomics Mapping Consortium (http://www.roadmapepigenomics.org/) was used to analyze the enhancer signal of different immune cell subpopulations. Cistrome database[44] (http://cistrome.org/db/#/) and ADASTRA database[57] (https://adastra.autosome.ru/susan) were used to analyze the binding proteins to rs2280381 in this study. GTEx[7] (https://gtexportal.org/home/) and ImmuNexUT database[70] (https://www.immunexut.org/) were used to analyze the eQTL result in this study. All other remaining data are available within the Article and Supplementary Files. Source data are provided with this paper.

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

## Acknowledgements

We thank Dr. Keishi Fujio and Dr. Mineto Ota at The University of Tokyo for their help to analyze the eQTL data. We thank Dr. Xiaoming Lu at Cincinnati Children's Hospital Medical Center for the help to perform MARIO analysis. This study was supported by grants from the National Natural Science Foundation of China 31930037 (N.S.), 31630021 (N.S.), 32141004 (N.S.) and 81102266 (Z.Y.), Shanghai Science and Technology Innovation Plan 21Y31900200 (N.S.), National Human Genetic Resources Sharing Service Platform 2005DKA21300 (N.S.), Shanghai municipal key medical center construction project 2017ZZ01024-002 (H.D.), Shenzhen Science and Technology Project JCYJ20180504170414637 (Z.Y.) and JCYJ20180302145033769 (Z.Y.), Shenzhen Futian Public Welfare Scientific Research Project FTWS2021006 (Z.Y.) and Sanming Project of Medicine in Shenzhen SZSM201602087 (N.S.).

## Author contributions

N.S., G.H. and T.Z. designed the project. T.Z., G.H., X.Z., Z.Y., Y.Z., Y.S., Z.H.Y. and N.X. performed the experiments. Y.Q. performed the ATAC-seq experiment. X.Z., Z.Y. and C.Y. analyzed the bioinformatics data. M.Z. and Y.G. analyzed the 4C-seq data. Y.W., W.Y., Y.T. and T.Z. analyzed the genetic data. J.B.H, B.N., K.M.K., L.C.K., and M.T.W. analyzed the genetic association data and revised the manuscript. H.D., X.L.Z., H.X. and J.M. collected human samples and performed analysis. N.S., G.H. and T.Z. prepared the manuscript.

## Competing interests

The authors declare no competing interests.
