## [Peer Review File · Nature Communications]

CRISPRa screen on a genetic risk locus shared by multiple autoimmune diseases identifies a dysfunctional enhancer that affects IRF8 expression through cooperative lncRNA and DNA methylation machineryReviewers' Comments:

Reviewer #1:

Remarks to the Author:

Zhou et al. identified risk allele of IRF8 and investigated molecular mechanisms how single nucleotide variant affects IRF8 expression in cell type specific manner in context to SLE. IRF8 is a key transcription factor of monocyte lineage differentiation and pro-inflammatory cytokine production in myeloid cells which are potentially pathogenic mechanism of lupus development. Risk alleles have identified by GWAS but their functional significance and regulatory mechanisms leading to difference in eQTL in immune cells are not fully understood. In this study, authors tried to investigate molecular mechanisms of IRF8 snp rs2280381 which is identified by CRISPR-based screening and this allele regulates IRF8 expression by recruitment of snp-recognizing transcription factor, epigenetic regulation, and non-coding enhancer RNA in this area. These finding was conducted meticulously by using cutting-edge technologies (CRISPR, ChIRP, FAIRE-qPCR, 4C sequencing etc). Findings are intriguing and translational significance in the field, too.

Although current research bears many valuable observations and technological advancement, there are several issues noted. Identification of risk snp (in figure 1) by CRIPSR and eQTL assay were concluded by the increased expression of IRF8 in monocytes. Indeed, this line of observation is consistent with the previous findings. Lack of IRF8 protects lupus-prone mice from the disease development. IRF8 expression is required for optimal induction of inflammatory function in myeloid cells. However, conclusion of study (summarized in figure 6) is opposite. Risk allele prevent binding of PU.1 and subsequent expression of long non-coding RNA and increased methylation of IRF8 promoter, leading to a decrease in IRF8 expression. One possible explanation is biased result from using U-937 cell line for most of mechanism studies. Although U-937 cell line replicates many phenotypic/functional characteristics of primary monocytes, it is well known that there are many intrinsic differences reside between them. Comparative analysis of gene expression (Darren M. Riddy et al 2018 PLOSONE) showed *Irf8* expression is very different in U-937 cell line compared to primary monocytes. Induction of IRF8 expression by stimulation is more dramatically different. This line of evidence suggests that genetic study from this cell line potentially generates non-physiological result.

Other points:

- Replica from cell line (in figure 2 and figure 5) cannot be used as individual samples.
- In figure 4, GAPDH was used as a negative control of RIP-qPCR. Since GAPDH resides in cytoplasm and only few are in nucleus, it is not a fair control. Please use other nucleus-resident RNA as a negative control.

Reviewer #2:

Remarks to the Author:

In this manuscript, Zhou and colleagues performed a CRISPRa screen on a genetic locus that is implicated in multiple autoimmune diseases. They identified the region around variant rs2280381 as a distal enhancer of IRF8 that interact with IRF8 promoter via DNA looping. Mechanistically, they proposed that rs2280381 controls the expression of enhancer RNA AC092723.1, which in turn recruits TET1 to IRF8 promoter leading to demethylation at the IRF8 locus. Furthermore, they also identified PU.1 as a key factor that binds to the rs2280381 locus and regulates AC092723.1 and IRF8 expression. The current study has made significant advance in revealing a novel molecular mechanism linking an autoimmune disease susceptible locus to IRF8 expression. The authors employed a number of cutting-edge technologies in their work, including CRISPRa screen, genome editing, FAIRE, 4C-sequencing, ChIRP, RIP, and DAPA. Their findings are likely to generate significant interests in the field. Despite its potential, the current manuscript has a few weaknesses that need to be addressed to strengthen its conclusions.

1. This study used ChIP-qPCR (Fig. 2E, Fig. 4J, Fig. 5D, 5E, 5G) to show histone modifications or transcription factor binding to a specific genomic locus. Although ChIP-qPCR has its value as a tool in pilot experiment, it is prone to biased observations due to the finicky nature of qPCR. The current standard in the field is to use ChIP-seq to show unbiased view of histone or transcription factor binding to DNA. Similarly, the ChIRP-qPCR experiment in Fig. 4D, 4E should be replaced by ChIRP-seq.
2. It is very interesting that rs2280381 locus interacts with IRF8 promoter through DNA looping (Fig. 2). Are there CTCF binding sites around IRF8 and rs2280381 to facilitate loop formation?
3. It is not clear why the authors switched from dCas9-VP64 to CRISPR-SAM in to study the consequence of activated rs2280381 locus in Fig. 3B,3C. Please provide rationale.
4. Fig. 3 showed loss-of-function study on AC092723.1's role in regulating IRF8 expression. What happens to IRF8 expression if AC092723.1 is overexpressed in a gain-of-function experiment?
5. Fig. 4F seems to suggest AC092723.1 contributes to loop formation. However, no evidence is provided. To prove this point, either Hi-C or 4C assay need to be performed using AC092723.1 KO cells?
6. In Fig. 4K, DNA methylation status should be presented at single nucleotide base level resolution. How many CpG sites are around IRF8 promoter, and what's the demethylation rate for each site?
7. Fig. 5 showed PU.1 binding to the rs2280381 region. Where are the putative PU.1 binding sites located in this region? Does the C to T conversion weaken PU.1 binding? Is it possible to set up a reporter assay using rs2280381 sequence variants to test PU.1's transcription activity?

Reviewer #3:

Remarks to the Author:

Nature Communications revision 309485

Cis-regulatory genetic variation is thought to be a key contributor to disease etiology and progression, but is understood at the mechanistic level at only few genomic loci. The authors present a strategy to dissect the function of autoimmune disease-associated genetic variants that relies on prioritising putative immune cell relevant SNPs in the IRF8 region, that are then targeted with dCas9-VP64 to identify functional enhancers controlling IRF8 expression.

This approach identifies rs2280381 as a putative causal variant in the IRF8 locus. rs2280381 is embedded in a monocyte-specific enhancer (open chromatin, H3K27ac), which displays looping interaction with the IRF8 promoter. Deletion of rs2280381-harboring region in a monocyte cell line (but interestingly not in a B or a T cell line) results in pronounced downregulation of IRF8 expression. The authors also identify a putative enhancer RNA AC092723.1 in the rs2280381-containing enhancer locus. Knockdown of this RNA leads to pronounced downregulation of IRF8 (but not vice versa), as does CRISPR mediated deletion of part of the AC092723.1 locus. ChIRP demonstrated that AC092723.1 interacts with IRF8 and rs2280381, indicating a possible role in enhancer-promoter loop formation. AC092723.1 interacts with TET1, and TET1 occupancy at the IRF8 promoter is reduced upon AC092723.1 knockdown, suggesting a mechanism whereby AC092723.1 mediates TET1 binding and hypomethylation at the IRF8 promoter. In addition to TET1, PU.1 also binds to the IRF8 promoter. By generating isogenic cell lines that carry different alleles of rs2280381, the authors demonstrate that the C/C non-risk allele is associated with higher chromatin accessibility, higher occupancy of enhancer marks and higher IRF8 and AC092723.1 expression than the T/T SLE risk allele of rs2280381.

Overall this is a timely and carefully executed and interpreted study with high quality data generated through the use of an impressive array of techniques to decipher the molecular mechanisms underlying a disease-associated non-coding genetic variant.

I have however some concerns that would need to be addressed before I could recommend this work for publication.

Major comments:

1.) I am not sure about the general approach of targeting dCas9-VP64 to enhancers to delineate their activity. This recruits a transcriptional activator to an already active regulatory region, which at least in some cases may not lead to further increase in target gene expression?

This would also explain the overall very modest effects seen in Figure 1D.

A much more common approach to identify functional enhancers is to interfere with enhancer function by dCas9-KRAB targeting (for example Fulco et al. Science 2016 (PMID: 27708057), and many studies since. Why have the authors chosen CRSIPRa over CRISPRi for functional enhancer delineation?

In the authors defense, a similar approach has been taken (Simeonov et al Nature 2017 PMID: 28854172) – but importantly the aim of this study was to identify stimulus-responsive enhancers in resting T cells.

Of note, Miguel-Escalada Nature Genetics 2019 (PMID: 31253982) use CRISPRa, CRISPRi and CRISPR mediated enhancer deletion for validation; this may be a good paper to compare the advantages/disadvantages of these approaches.

Overall, the approach the authors have taken needs to be discussed in the context of the existing literature, including the papers listed above.

2.) The statement that the rs2280381-containing region has the strongest regulatory effect among these SNPs is questionable based on the data in Figure 1D– it appears that the cited value could be partially driven by an outlier measurement in one of the three replicates? Other putative enhancers appear to show less variance/more significance in two tailed Student's t-test? I have to say though that on the balance of the other evidence presented in this paper, I am convinced that rs2280381-containing region is an enhancer of IRF8 – but I am not convinced that it is the strongest IRF8 enhancer in the locus based on the data in Figure 1D.

3.) Genes up- and downregulated upon deletion of IRF8 enhancer: more analyses required here. How many are likely to be direct targets of IRF8? Is there a chance that any of these genes may be directly regulated by the rs2280381-containing enhancer region?

4.) How was the DAPA experiment controlled? These kinds of experiments are notorious for high noise levels – did the authors include a control genomic locus?

5.) What is the effect of the C/C allele on TET1 binding in the IRF8/ rs2280381 locus?

Minor comments:

1.) lines 69/70 : sequence-specific DNA sequences ('sequence' duplicated)

2.) Line 95: little autoimmune disease risk variants...should be 'few'

3.) Figure 1B: should be dCas9-VP64 instead of dCas9-vp64

4.) Figure 1B (and also elsewhere in the manuscript): I would prefer ATAC-seq instead of

5.) 'Chromatin accessibility' – DNaseI HS assays and ATAC-seq both measure chromatin accessibility.

6.) Figure 2K-M: not clear to me how these experiments were done. I assume the quantification of IRF8 expression has been done on bulk cells, in which only a minority (in the case of monocytes) of the rs2280381 alleles have been edited? And those that have represent a mixture of different individual deletions, as shown in the supplemental data?

7.) Figure 3B-C: CRISPR SAM seems to be much more potent than dCas9-VP64 (Figure 1D) in increasing IRF8 expression when targeted to the rs2280381-containing enhancer. This is consistent with published literature which is not discussed, for example: Konermann et al Nature 2015 (PMID: 25494202)

8.) Figure 4A: some of the legend needs rephrasing ('Ultrasonic'; 'Combination') for more clarity.

9.) Line 324: detected is not wrong here but I think potentially misleading; 'analysed' or 'interrogated' is better.

10.) Can the authors include a reference for DAPA?

11.) Lines 490-494: "The accessibility of transcription factor binding sites is significantly heterogeneous in human immune cells, monocytes exhibited high activity of PU.1, and PU.1 is a key lineage-determining TF for priming monocyte-specific enhancers⁵², the binding of PU.1 to the rs2280381 locus may contribute to its function as a cell-type-specific enhancer."

I am not sure what the authors want to say here exactly, or if this sentence is complete. Possibly too much to fit into one sentence?

12.) Can the authors and discuss cite references that describe long enhancer RNAs and their possible roles in enhancer-promoter interactions?

13.) Please check and rephrase, where necessary, the wording especially in the methods section, for example in sentences like this (lines 751-754):

Enzyme was inactivated at 65 °C for 20 min, add 1.4 mL 10 x T4 DNA ligation buffer, 100 Units of T4 DNA ligase, supplement Milli-Q ddH₂O to 14 mL and ligate at room temperature for 4 h.

14.) Please rephrase this sentence (lines 499-501):

However, whether other proteins involved in this allele specific regulation and the mechanism forming cell-type-specific enhancer still deserve to be studied in more depth.

15.) Figure 6: DNA hypomethylation is better than DNA-unmethylation

16.) Line 848: oligos instead of oligoes

Point by point rebuttal to referees' comments

We thank the referees for their very thoughtful and constructive comments. We believe we have been able to address all of them, we have conducted our critical mechanistic experiments in human primary monocyte. Besides, we added plenty of sequencing data and immune subset eQTL data from a few authoritative databases to facilitate our conclusion. This and other major insights are described below. We believe this is a much-improved manuscript.

1. We combined genetic, epigenetic and CRISPRa screen successfully identified the functional variants in the *IRF8* locus. For the top 5 SNP-containing region, we further carried out CRISPR SAM activation assay and KRAB-dCas9 interference assay and validated that the rs2280381-containing region has the strongest regulatory function among these genetic variants.

2. Knockout the rs2280381-containing region significantly down-regulated *IRF8* expression in U-937 cells and had no impact in Raji or Jurkat cells, indicating rs2280381-containing region acts as a cell-type-specific enhancer. This observation was further convinced by disturbing rs2280381-containing region in human primary CD3⁺ T cells, CD19⁺ B cells and CD14⁺ monocytes.

3. The distal regulation of the rs2280381-containing region on *IRF8* depends on chromatin loop formation. We utilized circular chromatin conformation capture (4C) sequence assay to confirm the existence of interaction between the rs2280381-containing region and *IRF8* promoter. We also conducted a 4C assay on human primary monocyte and observed the interaction exists between the rs2280381-containing region and *IRF8* promoter, which is consistent with the results in U-937 cells.

4. We uncovered the complex components formed by rs2280381 enhancer and *IRF8* promoter, in which lncRNA *AC092723.1* has a critical role in loop conforming and recruiting.

- Knockdown *AC092723.1* significantly down-regulated *IRF8* expression

both in U-937 and human primary monocyte.

- We captured the RNA-DNA interaction through the CHIRP assay and found that *AC092723.1* interacts with the rs2280381-containing region and *IRF8* promoter in U-937 and primary monocyte.
- We conducted a 4C-seq assay to detect the role *AC092723.1* in chromatin loop formation. Knockdown *AC092723.1* in monocyte sharply decreased the interaction signal of the rs2280381-containing region and *IRF8* promoter.
- *AC092723.1* recruit demethylation enzyme TET1 to facilitate *IRF8* expression. We performed bioinformatic analysis, RIP assay, and ChIP assay, discovering that *AC092723.1* acts as a scaffold recruiting TET1 to the *IRF8* promoter. These experiments are both carried out in U-937 cells and primary monocytes, convincing our conclusion on *AC092723.1*.

5. The methylation level of the *IRF8* promoter can be increased by *AC092723.1* interference. We amplified 170 bp length fragment harboring 14 CPG sites and conducted Bisulfite Conversion PCR, and we found that knockdown *AC092723.1* increased the methylation level of *IRF8* promoter (control 6.55%, KD 14.88%).

6. To elucidate the allele-specific regulation of rs2280381 risk T allele and non-risk C allele on rs2280381 enhancer activity, we used a prime editing system to precisely edit rs2280381 variants and successfully generate clones harboring rs2280381 C homozygote, C/T heterozygote, T homozygote.

- rs2280381 risk T allele clones exhibit lower expression of *IRF8* than rs2280381 non-risk C allele clones, which is consistent with eQTL data in primary monocyte (ImmuNexUT).
- rs2280381 risk T allele enriched few H3K27ac signals than rs2280381 non-risk C allele analyzed by MARIO pipeline.
- Fragment-containing rs2280381 risk T allele has less chromatin accessibility than fragment-harboring rs2280381 non-risk C allele through FAIRE-AS-qPCR assay.

7. We combined DAPA-MS assay and Cistrome database data, revealing that PU.1 binds to the rs2280381-containing core sequence. Further, knockdown *PU.1* decreases the expression of *AC092723.1* and *IRF8* both in U-937 and primary monocyte. Moreover, allelic imbalance analysis in primary monocytes shows that the rs2280381 risk T allele has a weaker binding of PU.1 than the rs2280381 non-risk C allele (ADAstra database).

Reviewer #1 (Remarks to the Author):

Zhou et al. identified risk allele of IRF8 and investigated molecular mechanisms how single nucleotide variant affects IRF8 expression in cell type specific manner in context to SLE. IRF8 is a key transcription factor of monocyte lineage differentiation and pro-inflammatory cytokine production in myeloid cells which are potentially pathogenic mechanism of lupus development. Risk alleles have identified by GWAS but their functional significance and regulatory mechanisms leading to difference in eQTL in immune cells are not fully understood. In this study, authors tried to investigate molecular mechanisms of IRF8 snp rs2280381 which is identified by CRISPR-based screening and this allele regulates IRF8 expression by recruitment of snp-recognizing transcription factor, epigenetic regulation, and non-coding enhancer RNA in this area. These finding was conducted meticulously by using cutting-edge technologies (CRISPR, ChIRP, FAIRE-qPCR, 4C sequencing etc). Findings are intriguing and translational significance in the field, too.

Although current research bears many valuable observations and technological advancement, there are several issues noted. Identification of risk snp (in figure 1) by CRIPSR and eQTL assay were concluded by the increased expression of IRF8 in monocytes. Indeed, this line of observation is consistent with the previous findings. Lack of IRF8 protects lupus-prone mice from the disease development. IRF8 expression is required for optimal induction of inflammatory function in myeloid cells. However, conclusion of study (summarized in figure

6) is opposite. Risk allele prevent binding of PU.1 and subsequent expression of long non-coding RNA and increased methylation of IRF8 promoter, leading to a decrease in IRF8 expression. One possible explanation is biased result from using U-937 cell line for most of mechanism studies. Although U-937 cell line replicates many phenotypic/ functional characteristics of primary monocytes, it is well known that there are many intrinsic differences reside between them. Comparative analysis of gene expression (Darren M. Riddy et al 2018 PLOS ONE) showed *Irf8* expression is very different in U-937 cell line compared to primary monocytes. Induction of IRF8 expression by stimulation is more dramatically different. This line of evidence suggests that genetic study from this cell line potentially generates non-physiological result.

Response: We would appreciate the reviewer's recognitions on our integrated research approaches and translational significance in the field. This overall positive evaluation also encourages us to address all the critical concerns through doing challenging mechanistic experiments on primary monocytes to support our major conclusions.

First, we would clarify some points that we might not express or discuss clearly on the text. The reviewer points out that "Identification of risk snp (in figure 1) by CRIPSR and eQTL assay were concluded by **the increased expression** of IRF8 in monocytes. Indeed, this line of observation is consistent with the previous findings." Actually, both our study and eQTL analysis from different reliable datasets indicate that rs2280381 T risk allele **reduced the expression** of *IRF8* in monocytes (Fig. R1A-C).

Fig. R1. rs2280381 risk allele T is associated with low expression of *IRF8* in monocytes. (A) RT-qPCR detects the expression of *IRF8* in U-937 cell clones with rs2280381 different alleles generated by Prime editing. (B) eQTL data from GTEX. (C) eQTL data from ImmuNexUT project¹.

In our study, we first used dCas9-VP64 activation system targeting the rs2280381-containing region (Fig. R2A) and observed the up-regulated of *IRF8* expression (Fig. R2B), which demonstrating the function of enhancer harboring rs2280381 rather than the rs2280381 alleles.

To answer whether a genetic variant is a functional SNP regulating target gene expression, the only effective method is to construct cell clones harboring different alleles. We generated cell clones harboring different alleles demonstrating the rs2280381 T risk allele results in the low expression of *IRF8* in monocyte (Fig. R1A), this is consistent with the eQTL result from GTEX and ImmuNexUT project (Fig. R1B-C), especially the data from ImmuNexUT project, which provides the association between genetic variants and gene expression in different immune cell subsets. Of course, the direct evidence demonstrating the correlation between rs2280381 allele and *IRF8* expression is editing this SNP in primary monocytes. However, the precise editing of alleles without the byproducts such as fragment deletion or insertion in primary cells is still a big challenge.

For the role of *IRF8* in lupus development, Baccala *et al* reported that *IRF8* KO in NZB mice reduced the autoimmunity². In this study, the author used the *IRF8* whole-body knockout (KO) mouse model and found that *IRF8* KO results in the absence of pDCs and CD8 α + DCs and thus alleviates the lupus pathogenesis. As we all know, *IRF8* play an important role in the transcriptional regulation of hematopoiesis and peripheral immune responses, including monocyte, neutrophil, NK cells and dendritic cell (DC) lineage, commitment B cell development, germinal center reactions, T helper 1 (Th1) cell differentiation, and thymic selection³⁻⁶. Whole-body knockout (KO) *IRF8* would definitely alleviate many autoimmune diseases by impairing these cells development. Meanwhile, some studies using the conditional *IRF8* KO mouse revealed that *IRF8* may play a protective role in autoimmune disease. For instance, Pathak *et al* using B cell specific *IRF8* KO mouse found that mice deficient for *IRF8* produced anti-dsDNA Abs (Fig. R3), and the lack of *IRF8* in B cells leads to

Fig. R3

breakdown of B cell tolerance⁷, which could contribute to the development of autoimmune diseases. In addition, Ouyang *et al* using T cell specific *IRF8* KO mouse found that *IRF8* was induced steadily and inhibited TH17-cell

differentiation during TH17 lineage commitment (Fig. R4)⁸, as we all know, the abnormal of Fig. R4 TH17 cells contribute to the autoimmune disease development. Consistent with these findings, many RNA profiling data and our data show that *IRF8* expression is strongly decreased in SLE patients (Fig. R5).

Fig. R5. *IRF8* expression is down-regulated in SLE patients. (A) RT-qPCR detects the expression of *IRF8* in SLE patients. Samples are collected in our lab. (B-D) Public data from GEO indicates the low expression of *IRF8* in SLE patients.

These data support the notion that *IRF8* may play a protective role in SLE development. For the mechanism of low *IRF8* expression contributing to SLE pathogenesis, some studies may provide rational explanation. The impact of *IRF8* on the differentiation of DCs, monocytes and neutrophils are dose-dependent. Decreased expression of *IRF8* could induce the differentiation of pDCs while *IRF8* KO results in the absence of pDCs⁴ (Fig. R6A, Δ +56

of *IRF8*, *IRF8* expression would decrease after deletion of this enhancer), which are the most potent producers of type I IFNs strongly contribute to lupus pathogenesis⁹. Meanwhile, Low dose of *IRF8* also prompts the differentiation of Ly6C⁻ Mo, but high dose of *IRF8* has little effect on this cell type differentiation (Fig. R6B)⁴. More importantly, Ly6C⁻ Mo is the patrolling monocytes that promotes the pathogenesis of early lupus-like glomerulonephritis (Fig.R6C-D)¹⁰. Together, the role of *IRF8* in cell development and SLE is very complex. It is likely that the function of *IRF8* in SLE is stage specific or cell-type specific, which still need to be deciphered in future. For this part, we have added our comment in the discussion part of the revised manuscript.

Another problem that questioned by reviewer is that whether U-937 cells can act as the cell model to mimic the real regulation of rs2280381 on *IRF8* expression in primary monocyte. Actually, in our previous study, we observed that some cell lines cannot mimic the state of primary cells. For example, Raji

Fig. R7

B cells cannot fully mimic the miR-146a expression regulation as in primary B cells since the different epigenomic

modification in the two cells¹¹. So, what's the situation of rs2280381 site in U-937 cells? Analysis of epigenomic landscape of this site in U-937 cells and primary monocytes, we can observe the similar enhancer marker signal such as the high chromatin accessibility in both two cell types (Fig. R7). We also performed 4C-seq to compare the chromatin 3D structure in this site, as shown in Fig. R8A, the chromatin 3D structure in this region between two cell types are similar. More importantly, editing this region in U-937 cells and primary monocytes could both down-regulate *IRF8* expression (Fig. R8B-C).

Fig. R8. (A) The chromatin 3D structure of rs2280381 locus is similar between U-937 cells and primary monocyte. (B) Deletion of rs2280381-containing region reduced *IRF8* expression in U-937 cells. (C) Disruption of rs2280381-containing region decreased *IRF8* expression in primary monocyte.

To obtain more solid evidence to support our mechanistic conclusions on the regulatory function of the rs2280381 locus, we repeated all mechanistic studies with primary monocytes as suggested by the reviewer and obtained results consistent with what we found in U-937 cells. These new data have been presented in (Figure R9). We believe that these mechanistic experiments on primary monocytes will enhance the rigor of our novel findings and have been updated in the revised manuscript.

Fig. R9. The mechanism study in U-937 and primary monocyte. (A-B) Knockdown *AC092327.1* by ASO reduced *IRF8* expression both in primary monocyte and U-937 cells. **(C-D)** Knockdown *TET1* down-regulated *IRF8* expression both in primary monocyte and U-937 cells. **(E-F)** Knockdown *PU.1* decreased *IRF8* and *AC092327.1* expression both in primary monocyte and U-937 cells. **(G)** The distribution of *AC092327.1* in nucleus and cytoplasm location in primary monocyte and U-937 cells. **(H)** ChIRP experiment demonstrates that *AC092327.1* could bind with rs2280381-containing region and *IRF8* promoter region both in primary monocyte and U-937 cells. **(I)** RIP-qPCR assay indicates

the interaction between *TET1* and *AC092327.1* both in primary monocyte and U-937 cells. (J) ChIP-qPCR assay suggests *AC092327.1* KD impaired the *TET1* enrichment at *IRF8* promoter region both in primary monocyte and U-937 cells.

For the problem that “a study (Darren M. Riddy et al 2018 PLOS ONE) compared the *IRF8* expression in U-937 cell line and primary monocyte, the induction of *IRF8* is dramatically different”¹². We carefully read their work, they compared some genes’ expression in Monocyte-Like Cell Lines (THP1, HL60 and U-937) and CD14+ monocytes (CD14+ monocytes were named as PBMCs in this paper) after stimulation or differentiation. In their study, CD14+ monocytes were first treated with GM-CSF (GM-CSF was named as GC in the paper) for 6 days, and they found that GM-CSF treatment has no effect on *IRF8* expression (Fig. R10A and R10C). Then, GM-CSF treated CD14+ Monocytes

Fig. R10

(M(GC)) were further activated with LPS+IFN γ (M(GC)LPS/IFN γ), which significantly increased *IRF8* expression (Fig. R10A and R10C). In contrast, PMA treated U-937 cells decreased *IRF8* expression (Fig. R10B and R10D). These results are not surprising because 1) GM-CSF and PMA clearly will activate very different pathways as mentioned in

this paper (Darren M. Riddy et al 2018 PLOS ONE); 2) LPS and IFN γ have

been demonstrated as the effective inducers of *IRF8* expression in monocyte/macrophage, especially IFN- γ stimulation^{13,14}, but this paper has not detected the *IRF8* expression of U-937 cells treated with PMA+LPS+IFN γ (Fig. R10B); 3) although both stimuli could induce the macrophage-like phenotype, the state induced by two stimuli are different. Usually, PMA induced U-937 cells differentiation to M0 macrophage, but GM-CSF induced monocytes to M1-like macrophage¹⁵⁻¹⁸. We think the *IRF8* expression data should be interpreted with cautions based on comparing the *IRF8* expression in the different stimulation conditions for the two different cell types. We would appreciate the point that regulation of *IRF8* expression could be complicated on different conditions, but we do not think the data driven from this article are conflicted to our findings.

Other points:

- Replica from cell line (in figure 2 and figure 5) cannot be used as individual samples.

Response: In the two figures, cells were edited by CRISPR-Cas9 technology, and single cell was sorted into 96-well plate by FACS, after culturing for about 2 weeks, the genotype of each cell clone was identified. For each genotype, we chose several clones to test the effect (Fig.R11), so we think these clones are individual sample replicates. This is accepted by other publication¹⁹.

Fig. R11. Flow chart for generating genomic fragment deletion clones using the CRISPR-Cas9 technology.

- In figure 4, GAPDH was used as a negative control of RIP-qPCR. Since

GAPDH resides in cytoplasm and only few are in nucleus, it is not a fair control.
Please use other nucleus-resident RNA as a negative control.

Response: I think the experiment you point out is the ChIRP assay in Fig.4B-4E. In this assay, we have used *NEAT1*, a lncRNA predominately resides in nucleus, as control.

Reference

1. Ota, M. *et al.* Dynamic landscape of immune cell-specific gene regulation in immune-mediated diseases. *Cell* **184**, 3006-3021.e17 (2021).
2. Baccala, R. *et al.* Essential requirement for IRF8 and SLC15A4 implicates plasmacytoid dendritic cells in the pathogenesis of lupus. *Proc Natl Acad Sci U S A* **110**, 2940-5 (2013).
3. Adams, N.M. *et al.* Transcription Factor IRF8 Orchestrates the Adaptive Natural Killer Cell Response. *Immunity* **48**, 1172-1182.e6 (2018).
4. Murakami, K. *et al.* A RUNX-CBF β -driven enhancer directs the Irf8 dose-dependent lineage choice between DCs and monocytes. *Nat Immunol* **22**, 301-311 (2021).
5. Wang, H. *et al.* Transcription factors IRF8 and PU.1 are required for follicular B cell development and BCL6-driven germinal center responses. *Proc Natl Acad Sci U S A* **116**, 9511-9520 (2019).
6. Herzig, Y. *et al.* Transcriptional programs that control expression of the autoimmune regulator gene Aire. *Nat Immunol* **18**, 161-172 (2017).
7. Pathak, S., Ma, S., Shukla, V. & Lu, R. A role for IRF8 in B cell anergy. *J Immunol* **191**, 6222-30 (2013).
8. Ouyang, X. *et al.* Transcription factor IRF8 directs a silencing programme for TH17 cell differentiation. *Nat Commun* **2**, 314 (2011).
9. Rowland, S.L. *et al.* Early, transient depletion of plasmacytoid dendritic cells ameliorates autoimmunity in a lupus model. *J Exp Med* **211**, 1977-91 (2014).
10. Kuriakose, J. *et al.* Patrolling monocytes promote the pathogenesis of early lupus-like glomerulonephritis. *J Clin Invest* **129**, 2251-2265 (2019).
11. Hou, G. *et al.* SLE non-coding genetic risk variant determines the epigenetic dysfunction of an immune cell specific enhancer that controls disease-critical microRNA expression. *Nat Commun* **12**, 135 (2021).
12. Riddy, D.M. *et al.* Comparative genotypic and phenotypic analysis of human peripheral blood monocytes and surrogate monocyte-like cell lines commonly used in metabolic disease research. *PLoS One* **13**, e0197177 (2018).
13. Kantakamalakul, W. *et al.* Regulation of IFN consensus sequence binding protein expression in murine macrophages. *J Immunol* **162**, 7417-25 (1999).
14. Zhao, J. *et al.* IRF-8/interferon (IFN) consensus sequence-binding protein is involved in Toll-like receptor (TLR) signaling and contributes to the cross-talk between TLR and IFN-gamma signaling pathways. *J Biol Chem* **281**, 10073-80 (2006).

15. Buchacher, T., Ohradanova-Repic, A., Stockinger, H., Fischer, M.B. & Weber, V. M2 Polarization of Human Macrophages Favors Survival of the Intracellular Pathogen *Chlamydia pneumoniae*. *PLoS One* **10**, e0143593 (2015).
16. Bezold, V. *et al.* Glycation of macrophages induces expression of pro-inflammatory cytokines and reduces phagocytic efficiency. *Aging (Albany NY)* **11**, 5258-5275 (2019).
17. Gleissner, C.A. Macrophage Phenotype Modulation by CXCL4 in Atherosclerosis. *Front Physiol* **3**, 1 (2012).
18. Au - Mily, A. *et al.* Polarization of M1 and M2 Human Monocyte-Derived Cells and Analysis with Flow Cytometry upon *Mycobacterium tuberculosis* Infection. *JoVE*, e61807 (2020).
19. Gupta, R.M. *et al.* A Genetic Variant Associated with Five Vascular Diseases Is a Distal Regulator of Endothelin-1 Gene Expression. *Cell* **170**, 522-533.e15 (2017).

Reviewer #2 (Remarks to the Author):

In this manuscript, Zhou and colleagues performed a CRISPRa screen on a genetic locus that is implicated in multiple autoimmune diseases. They identified the region around variant rs2280381 as a distal enhancer of IRF8 that interact with IRF8 promoter via DNA looping. Mechanistically, they proposed that rs2280381 controls the expression of enhancer RNA AC092723.1, which in turn recruits TET1 to IRF8 promoter leading to demethylation at the IRF8 locus. Furthermore, they also identified PU.1 as a key factor that binds to the rs2280381 locus and regulates AC092723.1 and IRF8 expression. The current study has made significant advance in revealing a novel molecular mechanism linking an autoimmune disease susceptible locus to IRF8 expression. The authors employed a number of cutting-edge technologies in their work, including CRISPRa screen, genome editing, FAIRE, 4C-sequencing, ChIRP, RIP, and DAPA. Their findings are likely to generate significant interests in the field. Despite its potential, the current manuscript has a few weaknesses that need to be addressed to strengthen its conclusions.

1. This study used ChIP-qPCR (Fig. 2E, Fig. 4J, Fig. 5D, 5E, 5G) to show histone modifications or transcription factor binding to a specific genomic locus. Although ChIP-qPCR has its value as a tool in pilot experiment, it is prone to biased observations due to the finicky nature of qPCR. The current standard in the field is to use ChIP-seq to show unbiased view of histone or transcription factor binding to DNA. Similarly, the ChIRP-qPCR experiment in Fig. 4D, 4E should be replaced by ChIRP-seq.

Response: We thank the reviewer for this constructive suggestion. We agree that ChIP-seq can provide an unbiased and comprehensive view of the binding status of histone modification marks and transcription factors to genomic DNA. We believe that we can follow the strategy of many elegant

studies published recently to obtain unbiased and comprehensive epigenomic annotation of the rs2280381 locus. The strategy we adopted was to use sequencing data obtained from many reliable public databases (ENCODE, Cistrome) or from papers published in high-profile journals including Nature, Cell, Nature Genetics, etc., which provide many available and high-quality ChIP-seq from many cell lines and primary cell data sources. We analyzed these publicly available sequencing data and combined them with some of our own sequencing data to show unbiased information regarding to the binding status of histone modifications or transcription factors to the rs2280381 locus, and further validated these analyses using ChIP-qPCR. The data source used in this part is from the following papers: Nasser *et al*, Nature, PMID 33828297; Harley, *et al*. Nature genetics, PMID 29662164; Heinz S, et al. Cell. PMID 3014616; Schmidt SV, et al. Cell Res. PMID 26729620; Kang K, et al. Immunity. PMID 28813657 and Verma *et al*. Nat Genet. PMID 29203910.

For Fig. 2E and 2F, the ChIP-qPCR data and FAIRE-qPCR data of U-937 has been replaced with H3K27ac ChIP-seq data and ATAC-seq data by analyzing the public H3K27ac ChIP-seq data and ATAC-seq data of U-937 cells (Sequencing data is from Nasser *et al*, Nature, PMID 33828297) (Fig. R12A), these figures have been updated as Fig. 2E in the revised manuscript. Since H3K27ac signal is positively correlated with ATAC-seq, we also performed ATAC-seq assay to reveal the chromatin landscape of this region in primary monocytes. As shown in Fig. R12B, this region has high chromatin accessibility in primary monocytes, which is similar with U-937 cells.

Fig. R12. (A) ATAC-seq data and H3K27ac ChIP-seq data for the rs2280381 site in U-937 cells. (B) ATAC-seq analysis of the chromatin accessibility of rs2280381 site in primary monocytes.

For Fig. 5D, 5E, 5K, we used the MARIO¹ pipeline to analyze the allelic distribution of H3K27ac and PU.1 at the rs2280381 site using public ChIP-seq data. MARIO (Measurement of Allelic Ratios Informatics Operator) pipeline was developed by our group to identify allele-dependent protein binding by weighing imbalance between the number of sequencing reads for each allele of a given genetic variant, the total number of reads available at the variant, and the number and consistency of available experimental replicates (Harley, *et al.* Nature genetics, PMID 29662164). MARIO is an easy-to-use, modular tool that extends existing methods by (1) calculating a score that explicitly reflects reproducibility across experimental replicates; (2) reducing run-time via utilization of multiple computational cores; and (3) allowing the user to directly provide genotyping data as input. This method has been adopted in several important papers (Harley, *et al.* Nature genetics, PMID 29662164; Hou, *et al.* Nature Communications, PMID 33420081; Lu, *et al.* Nature

Communications, PMID 33712590)¹⁻³. We applied MARIO to ChIP-seq datasets performed in rs2280381 heterozygous cell lines and discovered that rs2280381 showed strong bias in the non-risk allele direction for H3K27ac and PU.1 signal (Fig. R13), these figures have been updated as Fig. 5D and 5J in the revised manuscript.

Fig. R13. MARIO¹ pipeline analyzed the allelic distribution of H3K27ac and PU.1 at the rs2280381 site (A) Genotype-dependent binding of H3K27ac for rs2280381. Results with MARIO ARS value >0.4 across ChIP-seq datasets are included. The X-axis indicates the preferred allele, along with a value indicating the strength of the allelic behavior, calculated as one minus the ratio of the weak to strong read counts (e.g., 0.5 indicates the strong allele has twice the reads of the weak allele). (B) PU.1 binds more strongly to the rs2280381 C non-risk allele, as analyzed determined by MARIO methods using the PU.1 ChIP-seq data in primary monocytes or macrophage provided by ADAstra database⁴.

For Fig. 5G, we analyzed the PU.1 ChIP-seq data in monocytes and macrophage from Cistrome project⁵ (<http://cistrome.org/>) (Source data are from Heinz S, et al. Cell. PMID: 3014616, Schmidt SV, et al. Cell Res. PMID: 26729620 and Kang K, et al. Immunity. PMID: 28813657) and found that rs2280381 site has strong PU.1 signal (Fig. R14). This figure has been

updated as Fig. 5F in the revised manuscript.

Fig. R14 ChIP-seq in monocyte and macrophage indicates the PU.1 binds to rs2280381 site.

For Fig.4D, 4E, actually, we once performed ChIRP-seq to detect the lncRNA binding with genomic region. As shown in Fig. R15, although the peak intensity at the promoter site is relatively weak, the interaction between lncRNA *AC092723.1* and *IRF8* promoter region could be definitely observed in the ChIRP-seq. And the weak signals in this assay may be caused by the relatively low abundance of this lncRNA expression. Based on the above data, we used the ChIRP-qPCR data to demonstrate the interaction of *AC092723.1* and *IRF8* promoter region.

Fig. R15 ChIRP-seq indicates that *AC092723.1* could interact with *IRF8* promoter region.

For Fig. 4J, we analyzed the ChIP-seq data from Cistrome project (<http://cistrome.org/>) (Sequencing data is from Verma *et al.* Nat Genet PMID 29203910) and found that TET1 could enrich at *IRF8* promoter region (Fig.

R16).

Fig. R16 Tet1 ChIP-seq suggests the TET1 could enrich in *IRF8* promoter region.

2. It is very interesting that rs2280381 locus interacts with IRF8 promoter through DNA looping (Fig. 2). Are there CTCF binding sites around IRF8 and rs2280381 to facilitate loop formation?

Response: We analyzed the CTCF-Seq data from Cistrome project (<http://cistrome.org/>) and ENCODE project, and found that rs2280381 locus and *IRF8* promoter region enriched strong CTCF signal in THP1 monocytes and GM12878 B cells, indicating the CTCF may facilitate the loop formation. But in Jurkat T cells and CD4+ T cells, we only observed the CTCF signal enriched in rs2280381 region but not in *IRF8* promoter region (Fig. R17), suggesting the CTCF binding in the two sites is cell-type-specific.

Fig. R17. CTCF ChIP-seq data in *IRF8* locus of different cell line.

3. It is not clear why the authors switched from dCas9-VP64 to CRISPR-SAM in to study the consequence of activated rs2280381 locus in Fig. 3B,3C. Please provide rationale.

Response: In Fig. 1D, we used the dCas9-VP64 system and synthesized sgRNAs to perform CRISPR screen (Fig. R18A). In this system, synthesized sgRNAs without modified scaffold and cannot recruit other activation proteins and exhibits modest levels of gene activation⁶ (Fig. R18C). In Fig. 3B and 3C, we used CRISPR SAM system to validate and study the consequence of activated rs2280381 locus. The CRISPR-SAM system is composed of three components: dCas9-VP64, sgRNA with modified scaffold and MS2-p65-HSF1 fusion protein (Fig. R18B). The modified sgRNA could recruit MS2-p65-HSF1 activation domains protein to further enhance the gene activation level^{6,7} (Fig. R18C). We established cell line stably expressing dCas9-VP64, MS2-p65-HSF1 fusion protein, and modified sgRNAs targeting rs2280381-region, which could induce higher levels of gene activation compared to dCas9-VP64 activation system. Although the CRISPR SAM system has stronger activation effect than dCas9-VP64 system, the results drawn from the two system are consistent.

Fig. R18 Comparison of dCas9-VP64 system and CRISPR-SAM system⁷. (A-B) Schematic of the dCas9-VP64 activator and CRISPR-SAM activator. (C) RT-qPCR analysis of 3 endogenous human genes expression using different activators.

4. Fig. 3 showed loss-of-function study on AC092723.1's role in regulating IRF8 expression. What happens to IRF8 expression if AC092723.1 is overexpressed in a gain-of-function experiment?

Response: Actually, we transiently transfected plasmid expressing AC092723.1 or constructed stable cell line overexpressing AC092723.1, and we cannot observe the obvious up-regulation of IRF8 (Fig. R19A-B). But when we use CRISPR-SAM system targeting the promoter region of AC092723.1 to increase the AC092723.1 expression *in cis*, IRF8 expression was strongly elevated (Fig. R19C-D). This result was not surprising, because many studies have demonstrated that expression of *cis*-regulation of lncRNA from

transfected vectors did not localize to the sites of their genomic counterpart regions or exert their roles in *cis*, resulting in the little effect of target genes (Fig. 19 E-H Figures from Xiang *et al.* Cell research. PMID: 24662484)^{8,9}.

Fig. R19. In cis over-expression of lncRNA could regulate lncRNA target gene expression. (A-B) Overexpression of *AC092723.1* using expression vector resulted in little effect on *IRF8* expression. (C-D) *In cis* activation of *AC092723.1* by CRISPR SAM up-regulated *IRF8* expression. (E-F) Overexpression of lncRNA *CCAT1-L* *in trans* in expression vector resulted in no apparent activation of *MYC* (Figures from PMID: 24662484). (G-H) *In cis* overexpression of lncRNA-*CCAT1-L* using TALEN technology enhanced *MYC* expression, as revealed by RT-qPCR (Figures from PMID: 24662484).

5. Fig. 4F seems to suggest *AC092723.1* contributes to loop formation. However, no evidence is provided. To prove this point, either Hi-C or 4C assay need to be performed using *AC092723.1* KO cells?

Response: We have performed 4C assay in monocytes to compare the interaction intensity between *IRF8* promoter and rs2280381 enhancer communication with or without *AC092723.1* knockdown, as shown in Fig. R20, knockdown *AC092723.1* significantly decreased the communication intensity of the two regions. And we have added this figure in the revised manuscript. This figure has been updated as Fig. 4G-H in the revised manuscript.

Fig. R20. Knockdown *AC092723.1* decreased the interaction intensity between *IRF8* promoter site and rs2280381 site. 4C-seq analysis of contact profiles of the *IRF8* promoter site and rs2280381 site in primary monocytes with or without *AC092723.1* knockdown using a 2 kb window size in the main trend subpanel. Red arrow heads indicate the view point position, and black arrow heads indicate the target position. Gray dots indicate normalized contact intensities. Heat map displays a set of medians of normalized contact intensities calculated at different window sizes.

6. In Fig. 4K, DNA methylation status should be presented at single nucleotide base level resolution. How many CpG sites are around *IRF8* promoter, and what's the demethylation rate for each site?

Response: Thanks for your suggestion, we have carried out Bisulfite sequencing PCR (BSP) assay to detect the methylation status of single nucleotide base. As shown in Fig. R21, *AC092723.1* knockdown increases the methylation level of *IRF8* promoter region, the methylation rate in the detected region was changed from 6.55% to 14.88%. And this figure has been updated in our revised manuscript.

Fig. R21. Knockdown *AC092723.1* increased the methylation level of *IRF8* promoter region. White dot indicates the unmethylated CpG sites and black dot indicates methylated CpG sites.

7. Fig. 5 showed PU.1 binding to the rs2280381 region. Where are the putative PU.1 binding sites located in this region? Does the C to T conversion weaken PU.1 binding? Is it possible to set up a reporter assay using rs2280381 sequence variants to test PU.1's transcription activity?

Response: We predicted the PU.1 binding site of rs2280381 fragment using JASPAR database, the result indicates that putative PU.1 binding sites

located the nearby region of rs2280381 but not overlapped with rs2280381 (Fig. R22A-B). However, several PU.1 ChIP-seq data suggested that rs2280381 site is enriched with PU.1 signal (Fig. R22C).

Fig. R22. Analysis of PU.1 binding to rs2280381 site. (A) PU.1 motif predicted by JARSPER database. (B) PU.1 relative binding site in rs2280381 containing region. (C) PU.1 ChIP-seq in primary monocyte and macrophage shows PU.1 overlapped with rs2280381 site.

For the C to T conversion influences PU.1 binding, we used ADAstra database⁴, an online tool analysis of allelic dosage-corrected allele-specific

human Transcription factor binding sites based on the ChIP-seq data, analyzed the allelic imbalance of rs2280381 binding to PU.1. As shown in Fig. R23, rs2280381 reference allele C enriched 44 read counts and 19 read counts in two rs2280381 heterozygous macrophage cells, respectively. In contrast, allele T only enriched 8 read counts in both cells. These data reveal that C to T conversion weaken the PU.1 binding.

rs2280381 C>T in macrophages from peripheral blood

GTRD experiment ID	Uniprot ID	Ref read counts	Alt read counts	Estimated BAD	P-value Ref ↑	P-value Alt
EXP034643	SPI1_HUMAN	44	8	5/2	$2.5 \cdot 10^{-3}$	1.00
EXP034644	SPI1_HUMAN	19	8	5/2	0.32	0.43

Fig. R23. PU.1 ChIP-seq data from two rs2280381 heterozygous macrophage cells was used to analysis the allelic imbalance of PU.1 at rs2280381 site. C is the rs2280381 reference allele, T is the rs2280381 alternative allele. Read counts indicates the binding ability of PU.1 to rs2280381 different alleles. SPI1, PU.1.

Meanwhile, we performed the reporter assay suggested by the reviewer, we amplified about 200 bp fragment harboring the rs2280381 and cloned this fragment into the PGL3 basic vector containing 2000-bp *IRF8* promoter to study the enhancer activity of rs2280381 fragment (Fig. R24A). We found that fragment-containing rs2280381 C allele could slightly elevate the luciferase expression (Fig. R24B). In contrast, fragment-containing rs2280381 T allele has no effect on luciferase expression (Fig. R24B). After co-transfection with PU.1, the activity of fragment-containing rs2280381 C allele strongly increased, however, the activity of fragment-containing T allele was still not influenced (Fig. R24B). Luciferase report assay has been widely applied to study the enhancer function due to its convenience and high-throughput character. However, this assay cannot fully mimic the character of enhancer

since distal enhancers usually form complex 3D genomic structure with target gene promoter and enriched with histone modifications and transcription factors complex. In addition, the biological contexts (cell-type, stimulation, etc.) and fragment length cloned into the vector may also influenced the results obtained from luciferase reporter assay¹⁰. As the advent of CRISPR/Cas9-mediated sequence perturbation, this method has been considered the gold standard for identifying functional regulatory elements. Our study that integrated the CRISPR-mediated deletion, HDR, activation and interference has definitely identified rs2280381-containing region as a functional enhancer regulating *IRF8* expression, and rs2280381 alleles could differentially regulated *IRF8* expression. Meanwhile, the unbiased ChIP-seq data analysis using rs2280381 heterozygous macrophage cells also reveals PU.1 prefers binding to the allele C relative to allele T (Fig. R23).

Fig. R24. Luciferase reporter assay detect the allele-specific regulation of rs2280381. (A) The plasmid map of rs2280381 enhancer-*IRF8* promoter-pGL3 based luciferase report vector. (B) Luciferase report assay in HEK-293T with or without PU.1 over-expression.

Reference

1. Harley, J.B. *et al.* Transcription factors operate across disease loci, with EBNA2 implicated

- in autoimmunity. *Nat Genet* **50**, 699-707 (2018).
2. Lu, X. *et al.* Global discovery of lupus genetic risk variant allelic enhancer activity. *Nat Commun* **12**, 1611 (2021).
 3. Hou, G. *et al.* SLE non-coding genetic risk variant determines the epigenetic dysfunction of an immune cell specific enhancer that controls disease-critical microRNA expression. *Nat Commun* **12**, 135 (2021).
 4. Abramov, S. *et al.* Landscape of allele-specific transcription factor binding in the human genome. *Nat Commun* **12**, 2751 (2021).
 5. Liu, T. *et al.* Cistrome: an integrative platform for transcriptional regulation studies. *Genome Biol* **12**, R83 (2011).
 6. Konermann, S. *et al.* Genome-scale transcriptional activation by an engineered CRISPR-Cas9 complex. *Nature* **517**, 583-8 (2015).
 7. Chavez, A. *et al.* Comparison of Cas9 activators in multiple species. *Nat Methods* (2016).
 8. Xiang, J.F. *et al.* Human colorectal cancer-specific CCAT1-L lncRNA regulates long-range chromatin interactions at the MYC locus. *Cell Res* **24**, 513-31 (2014).
 9. Zhang, Y. *et al.* Circular intronic long noncoding RNAs. *Mol Cell* **51**, 792-806 (2013).
 10. Mulvey, B., Lagunas, T., Jr. & Dougherty, J.D. Massively Parallel Reporter Assays: Defining Functional Psychiatric Genetic Variants Across Biological Contexts. *Biol Psychiatry* **89**, 76-89 (2021).

Reviewer #3 (Remarks to the Author):

Nature Communications revision 309485

Cis-regulatory genetic variation is thought to be a key contributor to disease etiology and progression, but is understood at the mechanistic level at only few genomic loci. The authors present a strategy to dissect the function of autoimmune disease-associated genetic variants that relies on prioritising putative immune cell relevant SNPs in the IRF8 region, that are then targeted with dCas9-VP64 to identify functional enhancers controlling IRF8 expression. This approach identifies rs2280381 as a putative causal variant in the IRF8 locus. rs2280381 is embedded in a monocyte-specific enhancer (open chromatin, H3K27ac), which displays looping interaction with the IRF8 promoter. Deletion of rs2280381-harboring region in a monocyte cell line (but interestingly not in a B or a T cell line) results in pronounced downregulation of IRF8 expression.

The authors also identify a putative enhancer RNA AC092723.1 in the rs2280381-containing enhancer locus. Knockdown of this RNA leads to pronounced downregulation of IRF8 (but not vice versa), as does CRISPR mediated deletion of part of the AC092723.1 locus. ChIRP demonstrated that AC092723.1 interacts with IRF8 and rs2280381, indicating a possible role in enhancer-promoter loop formation. AC092723.1 interacts with TET1, and TET1 occupancy at the IRF8 promoter is reduced upon AC092723.1 knockdown, suggesting a mechanism whereby AC092723.1 mediates TET1 binding and hypomethylation at the IRF8 promoter. In addition to TET1, PU.1 also binds to the IRF8 promoter. By generating isogenic cell lines that carry different alleles of rs2280381, the authors demonstrate that the C/C non-risk allele is associated with higher chromatin accessibility, higher occupancy of enhancer marks and higher IRF8 and AC092723.1 expression than the T/T SLE risk allele of rs2280381.

Overall this is a timely and carefully executed and interpreted study with high quality data generated through the use of an impressive array of techniques to decipher the molecular mechanisms underlying a disease-associated non-coding genetic variant.

I have however some concerns that would need to be addressed before I could recommend this work for publication.

Major comments:

1.) I am not sure about the general approach of targeting dCas9-VP64 to enhancers to delineate their activity. This recruits a transcriptional activator to an already active regulatory region, which at least in some cases may not lead to further increase in target gene expression? This would also explain the overall very modest effects seen in Figure 1D.

Response: dCas9-VP64 activation is generally thought of as the “first generation” CRISPR activator. It depends on transcriptional activator VP64 to activate the gene expression. Compared with other CRISPR activation systems, it requires a relatively simple construct and exhibits modest levels of gene activation¹ (Fig. R25A-B). To further confirm our screen result in Figure 1D, we also adopted the CRISPR SAM activation assay and KRAB-dCas9 interference assay to validate the regulatory effect of top 5 genetic variants-harboring region and observed the similar result (Fig. R25C-F). This figure has been updated in our revised manuscript

Fig. R25. Tests of CRISPR activators on endogenous genes. (A) Schematic of the different activators¹. (B) ASCL1 and NEUROD1 expression after activated by different activators in HEK293T cells¹. (C) The components of CRISPR SAM activation system. (D) RT-qPCR analysis of *IRF8* expression in the CRISPR SAM activation experiment (n = 3, biological replicates). (E) The components of KRAB-dCas9 system. (F) RT-qPCR analysis of *IRF8* expression in the KRAB-dCas9 interference experiment (n = 3, biological replicates). Data are represented as mean \pm SEM and P-values are calculated using an unpaired two tailed Student's t-test.

A much more common approach to identify functional enhancers is to interfere with enhancer function by dCas9-KRAB targeting (for example Fulco et al. Science 2016 (PMID: 27708057), and many studies since. Why have the authors chosen CRISPRa over CRISPRi for functional enhancer delineation? In the authors defense, a similar approach has been taken (Simeonov et al Nature 2017 PMID: 28854172) – but importantly the aim of this study was to identify stimulus-responsive enhancers in resting T cells. Of note, Miguel-Escalada Nature Genetics 2019 (PMID: 31253982) use CRISPRa, CRISPRi and CRISPR mediated enhancer deletion for validation; this may be a good paper to compare the advantages/disadvantages of these approaches. Overall, the approach the authors have taken needs to be discussed in the context of the existing literature, including the papers listed above.

Response: Yes, you are right, more studies adopted dCas9-KRAB technology to identify the functional enhancers in the literature. Before starting this experiment, we once considered adopting the same strategy. However, in our study, the distance among several SNPs is less than 500 bp, and several studies reported that dCas9-KRAB system could induce histone modifications associated with heterochromatin across about 1 kb window around a sgRNA's target site²⁻⁴, which could provide the misleading information of the regulatory

effect of these SNP-containing region. For dCas9-VP64 system, this system depends on the VP64 proteins rather than induction histone modification to activate target gene expression. Based on this information, we finally choose dCas9-VP64 CRISPRa system to screen in our study. In our revised manuscript, we also performed the CRISPRi assay for the top 5 candidate SNP-containing region and updated this figure (Fig. R25E-F), Meanwhile, we have discussed these problems and cited these papers that you suggested. Thanks for your suggestion.

2.) The statement that the rs2280381-containing region has the strongest regulatory effect among these SNPs is questionable based on the data in Figure 1D– it appears that the cited value could be partially driven by an outlier measurement in one of the three replicates? Other putative enhancers appear to show less variance/more significance in two tailed Student's t-test? I have to say though that on the balance of the other evidence presented in this paper, I am convinced that rs2280381-containing region is an enhancer of *IRF8* – but I am not convinced that it is the strongest *IRF8* enhancer in the locus based on the data in Figure 1D.

Response: Thank you for pointing out this. To confirm the screen result of dCas9-VP64, for the top 5 SNP-containing region in the Figure 1D, we further carried out CRISPR SAM activation assay and KRAB-dCas9 interference assay to validate the regulatory effect of these SNP-containing regions. As shown in Fig. R26A-D, rs2280381-containing region also has the strongest regulatory effect on *IRF8* expression in both assays. Moreover, our mechanism studies support the rs2280381 locus as a functional enhancer regulating *IRF8* expression. We have updated these figures in the revised manuscript.

Fig. R26. CRISPR SAM and KRAB-dCas9 system validate the regulatory effect of SNP-containing region. (A) The components of CRISPR SAM activation system. (B) RT-qPCR analysis of *IRF8* expression in the CRISPR SAM activation experiment (n = 3, biological replicates). (C) The components of KRAB-dCas9 system. (D) RT-qPCR analysis of *IRF8* expression in the KRAB-dCas9 interference experiment (n = 3, biological replicates). Data are represented as mean \pm SEM and P-values are calculated using an unpaired two tailed Student's t-test.

3.) Genes up- and downregulated upon deletion of *IRF8* enhancer: more analyses required here. How many are likely to be direct targets of *IRF8*? Is there a chance that any of these genes may be directly regulated by the rs2280381-containing enhancer region?

Response: Thanks for your suggestion. Enhancers usually form gene loop with the promoter site to regulate target gene expression. To explore the genes directly modulated by rs2280381-containing region, we first analyzed the 4C-seq data from the rs2280381 view point and find that 66 genes' promoter regions exist interaction with rs2280381 site. Analysis of these genes' expression between WT and rs2280381 KO cell clones (38 genes' expression can be detected in RNA sequencing, remaining genes' expression cannot be detected), we found that the expressions of lncRNA *AC092723.1* and *IRF8* were significantly different between WT and rs2280381 KO cell clones (cutoff value: log₂ fold-change ≥ 1 and FDR < 0.05) (Table 1). This

Table 1

gene_id	Log ₂ fold change	FDR
AC092723.1	-1.7854	0.0285
IRF8	-1.0216	0.0302
RP11-482M8.1	-1.3134	0.2516
KIAA0513	0.4808	0.2630
MBTPS1	0.2247	0.2756
KLHDC4	-0.2315	0.2767
PABPN1L	-1.2277	0.3583
MAP1LC3B	0.1105	0.5761
AC010536.1	-0.4053	0.6301
GSE1	-0.0666	0.6409
COX4I1	-0.2899	0.7455
VPS9D1-AS1	-0.5176	0.8059
C16orf95	-0.3884	0.8515
SPATA33	-0.1177	0.8695
GCSH	-0.1845	0.8732
EMC8	0.0332	0.9031
LINC02132	-0.2998	0.9066
MTHFSD	0.0200	0.9210
RNU1-103P	0.1991	0.9426
CHMP1A	-0.0266	0.9438
AC040169.1	0.0881	0.9472
SNAI3	-0.0447	0.9580
RN7SL381P	-0.3264	0.9643
VPS9D1	0.1927	0.9646
BANP	0.0129	0.9652
RP4-536B24.4	-0.0415	0.9652
GINS2	-0.0732	0.9652
FBXO31	-0.1369	0.9703
MIR5189	-1.1192	0.9901
APRT	-0.1168	0.9911
LINC02181	1.0880	0.9912
CTD-2555A7.1	-0.2495	0.9912
KLHL36	-0.0156	0.9943
GAN	-0.0775	0.9973
MIR1910	0.1418	0.9997
LINC00917	3.1534	NA
CRISPLD2	-0.0244	NA
FOXC2	-1.8947	NA

comprehensive and stringent analysis provide further evidence supporting the conclusion that rs2280381-containing enhancer directly regulates the expression of *IRF8* and *AC092723.1*. Next, we adopted the Ingenuity Pathway Analysis (IPA) to predict the possible targets of *IRF8*, meanwhile, *IRF8* ChIP-seq data in monocytes was analyzed to find whether *IRF8* could bind to the promoter region of predicted genes, the genes' promoter region enriched with *IRF8* binding were considered as *IRF8* direct target. Finally, we got about 61 gene candidates, these genes include interferon inducible genes (ISG) (*ISG15*, *IFIT1*, etc.), immune response associated genes (*BANK1*, *CIITA*, *ICAM*, etc.), chemokines (*CCL22*, *CCL5*, etc.) and so on (Fig. R27). Analysis of RNA-sequencing data indicates that some of these genes' expression was up-

regulated or down-regulated after rs2280381 fragment was knockout (Fig. R27). For some genes, we performed RT-qPCR to validate their expression in WT and KO group, and observed the similar expression change pattern as RNA sequencing data (Fig. R28). Notably, *IRF8* negatively regulated genes such as *ISG15*⁵ and *IFIT1*⁶ or *IRF8* positively regulated genes including *CIITA*⁷ has been validated by other groups' studies. This part has been updated in the revised

manuscript.

Fig. R28. RT-qPCR validates *IRF8* target gene expression.

4.) How was the DAPA experiment controlled? These kinds of experiments are notorious for high noise levels – did the authors include a control genomic locus?

Response: Thank you for your suggestion. We have repeated this assay and include a random sequence as control (Supplementary Data Set 3). For choosing candidates, we set some criteria to filter the candidate proteins. The criteria are as follows: 1. proteins must be detected only in experiment group; 2. proteins should be transcription factors since our group found transcription factors (TFs) occupy multiple loci associated with complex genetic disorders⁸; (3) Proteins should be overlapped with ChIP-seq data in this site provided by Cistrome project⁹. For this part, we have expanded the method in the revised manuscript

5.) What is the effect of the C/C allele on TET1 binding in the *IRF8*/ rs2280381 locus?

Response: According to the reviewer's suggestion, we conducted related experiments in cell clones with T/T alleles and C/C alleles, but it was not ideal. We could not find statistically significant differences in rs2280381 alleles regulating TET1 enrichment in the *IRF8* promoter region (Fig. R29A). If we carefully exam the detailed molecular mechanisms underlying the action of this genetic variant, this unideal experimental result can be reasonably interpreted. In our study, we demonstrated that the rs2280381 allele directly and differently regulates the expression of lncRNA *AC092723.1*, and lncRNA functions as a scaffold recruiting TET1 to the *IRF8* promoter region but not the rs2280381 site (Fig. R29B). The distance between *IRF8* promoter and rs2280381 is about 64 kb, and alleles only can indirectly affect the TET1 binding in *IRF8* promoter, the current ChIP-qPCR method may not discriminate this tiny difference in a few samples (Fig. R29B), and establishment of the statistic significant association

between rs2280381 different alleles and TET1 enrichment in *IRF8* promoter may require hundreds of samples like the eQTL assay that finds the association between alleles and gene expression (Fig. R29C), which is a very huge and challenging task. If novel method was developed, we would be happy to try.

Fig. R29. (A) ChIP-qPCR detects the TET1 enrichment at *IRF8* promoter region using rs2280381 T/T or C/C clones. (B) Flow scheme of detecting the TET1 enrichment at *IRF8* promoter region using rs2280381 T/T or C/C clones. (C) Flow scheme of eQTL assay revealing the influence of rs2280381 alleles on TET1 enrichment at *IRF8* promoter region.

Minor comments:

1.) lines 69/70: sequence-specific DNA sequences ('sequence' duplicated)

Response: Thank you for pointing this out. We have revised in the manuscript.

2.) Line 95: little autoimmune disease risk variants...should be 'few'

Response: Thank you for pointing this out. We have modified this word in the manuscript.

3.) Figure 1B: should be dCas9-VP64 instead of dCas9-vp64.

Response: Thank you for pointing this out. We have used dCas9-VP64 instead of dCas9-vp64 in the Figure 1B of the revised manuscript.

4.) Figure 1B (and also elsewhere in the manuscript): I would prefer ATAC-seq instead of

5.) 'Chromatin accessibility' – DNaseI HS assays and ATAC-seq both measure chromatin accessibility.

Response: Thank you for pointing this out. We have used ATAC-seq instead of Chromatin accessibility in the revised manuscript.

6.) Figure 2K-M: not clear to me how these experiments were done. I assume the quantification of *IRF8* expression has been done on bulk cells, in which only a minority (in the case of monocytes) of the rs2280381 alleles have been edited? And those that have represent a mixture of different individual deletions, as shown in the supplemental data?

Response: We have added a flow scheme in the supplementary figure to describe the procedure of these experiments. For primary cells, it is difficult to isolate cells with consistent genotype to detect the downstream effect, so the quantification of *IRF8* expression was detected on bulk cells and the results represent a mixture of different individual deletions (Fig. R30A). Meanwhile, to

evaluate the editing efficiency in the edited cells, we first amplified the targeting locus harboring the editing region by specific primers. Then PCR products were sequenced by Sanger sequence, and the data of Sanger sequencing was analyzed with ICE (Inference of CRISPR Edit) (<https://ice.synthego.com/#/>), which can quantify the editing outcome observed in the mixed Sanger read and the analyzed results correlate well with next-generation sequencing of amplicons (Amp-Seq)¹⁰ (Fig. R30B).

Fig. R30. (A) Flow scheme of CRISPR-mediated fragment disruption in isolated primary cells. (B) ICE analysis flow scheme and the editing efficiency evaluated by ICE.

7.) Figure 3B-C: CRISPR SAM seems to be much more potent than dCas9-VP64 (Figure 1D) in increasing IRF8 expression when targeted to the rs2280381-containing enhancer. This is consistent with published literature which is not discussed, for example: Konermann et al Nature 2015 (PMID: 25494202)

Response: Yes, you are right. The CRISPR-SAM system is composed of three components: dCas9-VP64, sgRNA with modified scaffold and MS2-p65-HSF1 fusion protein. The modified sgRNA could recruit MS2-p65-HSF1 activation domains protein to further enhance the gene activation level^{1,11} (Fig. R25A). In Fig.1, we synthesized sgRNAs to perform CRISPR screen, and synthesized sgRNAs without modified scaffold, which cannot recruit additional activation domain resulting the modest levels of gene activation. In Fig.3B and 3C, we established cell line stably expressing dCas9-VP64, MS2-p65-HSF1 fusion protein and modified sgRNAs recruit additional activation domain targeting rs2280381-region, which could induce higher levels of gene activation compared to dCas9-VP64 activation system. In addition, we have cited this reference in our revised manuscript.

8.) Figure 4A: some of the legend needs rephrasing ('Ultrasonic'; 'Combination') for more clarity.

Response: Thanks for your suggestion. We have revised these words in the manuscript.

9.) Line 324: detected is not wrong here but I think potentially misleading;

'analysed' or 'interrogated' is better.

Response: Thanks for your suggestion. We have revised in the manuscript.

10.) Can the authors include a reference for DAPA?

Response: Thanks for your suggestion. We have repeated this experiment and included a reference for DAPA assay, and new data were added in the revised manuscript.

11.) Lines 490-494: "The accessibility of transcription factor binding sites is significantly heterogeneous in human immune cells, monocytes exhibited high activity of PU.1, and PU.1 is a key lineage-determining TF for priming monocyte-specific enhancers⁵², the binding of PU.1 to the rs2280381 locus may contribute to its function as a cell-type-specific enhancer."

I am not sure what the authors want to say here exactly, or if this sentence is complete. Possibly too much to fit into one sentence?

Response: Thank you for pointing out this. We have revised this sentence in the manuscript.

12.) Can the authors and discuss cite references that describe long enhancer RNAs and their possible roles in enhancer-promoter interactions?

Response: Thanks for your suggestion. We have discussed the role of long enhancer RNAs in enhancer-promoter interactions. Meanwhile, in our revised manuscript, we knockdown *AC092723.1* by ASO and detected the interaction intensity between *IRF8* promoter site and rs2280381 site, we found that knockdown *AC092723.1* could effectively reduce the interaction intensity between two sites, suggesting *AC092723.1* facilitates the loop formation, this provides direct evidence that lncRNA regulate chromatin structure.

13.) Please check and rephrase, where necessary, the wording especially in the methods section, for example in sentences like this (lines 751-754):

Enzyme was inactivated at 65 °C for 20 min, add 1.4 mL 10 x T4 DNA ligation buffer, 100 Units of T4 DNA ligase, supplement Milli-Q ddH₂O to 14 mL and ligate at room temperature for 4 h.

Response: Thank you for pointing this out. We have revised these words in the manuscript.

14.) Please rephrase this sentence (lines 499-501):

However, whether other proteins involved in this allele specific regulation and the mechanism forming cell-type-specific enhancer still deserve to be studied in more depth.

Response: Thanks for your suggestion, we have rephrased this sentence in the revised manuscript.

15.) Figure 6: DNA hypomethylation is better than DNA-unmethylation

Response: Thank you for pointing this out. We have revised these words in the manuscript.

16.) Line 848: oligos instead of oligoes

Response: Thank you for pointing this out. We have revised this word in the manuscript.

Reference

1. Chavez, A. *et al.* Comparison of Cas9 activators in multiple species. *Nat Methods* (2016).
2. Thakore, P.I. *et al.* Highly specific epigenome editing by CRISPR-Cas9 repressors for

- silencing of distal regulatory elements. *Nat Methods* **12**, 1143-9 (2015).
3. Gasperini, M. *et al.* A Genome-wide Framework for Mapping Gene Regulation via Cellular Genetic Screens. *Cell* **176**, 377-390.e19 (2019).
 4. Hollerer, I. *et al.* Evidence for an Integrated Gene Repression Mechanism Based on mRNA Isoform Toggling in Human Cells. *G3 (Bethesda)* **9**, 1045-1053 (2019).
 5. Salem, S. *et al.* Functional characterization of the human dendritic cell immunodeficiency associated with the IRF8(K108E) mutation. *Blood* **124**, 1894-904 (2014).
 6. White, C.L., Kessler, P.M., Dickerman, B.K., Ozato, K. & Sen, G.C. Interferon Regulatory Factor 8 (IRF8) Impairs Induction of Interferon Induced with Tetratricopeptide Repeat Motif (IFIT) Gene Family Members. *J Biol Chem* **291**, 13535-45 (2016).
 7. Marquis, J.F. *et al.* Interferon regulatory factor 8 regulates pathways for antigen presentation in myeloid cells and during tuberculosis. *PLoS Genet* **7**, e1002097 (2011).
 8. Harley, J.B. *et al.* Transcription factors operate across disease loci, with EBNA2 implicated in autoimmunity. *Nat Genet* **50**, 699-707 (2018).
 9. Liu, T. *et al.* Cistrome: an integrative platform for transcriptional regulation studies. *Genome Biol* **12**, R83 (2011).
 10. Hsiao, T. *et al.* Inference of CRISPR Edits from Sanger Trace Data. *bioRxiv*, 251082 (2018).
 11. Konermann, S. *et al.* Genome-scale transcriptional activation by an engineered CRISPR-Cas9 complex. *Nature* **517**, 583-8 (2015).

Reviewers' Comments:

Reviewer #1:

Remarks to the Author:

Author addressed all of critiques raised in the previous review.

Reviewer #2:

Remarks to the Author:

Zhou and colleagues have made extensive revisions to address the concerns raised in their original manuscript. They performed a suite of new experiments including ChIRP-seq, CRISPR-SAM, 4C assay, bisulfite-seq, and luciferase assays. Additionally, they conducted new analysis of their own data and publicly available data to address reviewers' questions. The current manuscript is strengthened significantly. I only have a minor suggestion. The luciferase reporter assay results in Fig. R24 are complementary to the data shown in Fig. 5. I feel it should be included in that figure. Otherwise, I congratulate the authors for a great study.

Reviewer #3:

Remarks to the Author:

The authors have done a really good job to answer my questions and importantly, they have included additional experimental data (CRISPRi) which have improved the quality of the manuscript and further strengthen their conclusions. I recommend this manuscript for publication; however, having gone through the changes in the manuscript, there are many passages where the quality of the data is not matched by the language used. This is a shame and I would urge the authors to enlist the help from a professional editor to bring the writing up to the level that their work deserves to be described in. As a non-native English speaker, I can completely relate to the difficulties the authors are experiencing here. Having faced similar issues, I have used professional editors in the past and it has definitely helped to improve the quality of my manuscripts. I strongly recommend the authors do the same for this manuscript.

I am outlining a few examples below where I feel the writing could be improved; however there are more and the entire manuscript would benefit from an overall review by a professional.

1. Lines 253-275: Enhancers usually form gene loop with the promoter site to regulate target gene expression. To explore the genes directly modulated by rs2280381-containing region, we first analyzed the 4C-seq data from the rs2280381 view point and find that 66 genes' promoter regions exist interaction with the rs2280381 site (Supplementary Data Set 2B). Analysis of these genes' expression between WT and rs2280381 KO cell clones, we found that the expressions of lncRNA AC092723.1 and IRF8 were significantly different between the two groups (\log_2 fold-change ≥ 1 and $FDR < 0.05$) (Supplementary Data Set 2C), suggesting AC092723.1 and IRF8 may be directly regulated by the rs2280381-containing enhancer region. After demonstrating IRF8 as a direct target of rs2280381-containing region, we adopted the Ingenuity Pathway Analysis (IPA) to predict the possible targets of IRF8. Meanwhile, IRF8 ChIP-seq data in monocytes (<http://cistrome.org/db/#/>)48,49 was analyzed to find whether IRF8 could bind to the promoter region of predicted genes, the genes' promoter region enriched with IRF8 binding were considered as IRF8 direct target. Finally, we got about 61 gene candidates (Supplementary Data Set 2D). For some genes, such as ISG15, IFIT1 and CIITA, we performed RT-qPCR to validate their expression. Consistent with the previous reports, IRF8 could negatively regulated ISG1550 and IFIT136 expression and positively regulated CIITA51 expression in our data (Fig. S3B-D). Next, differentially expressed genes were analyzed using gene ontology (GO) analysis and revealed that these genes are highly enriched in expected biological process such as inflammatory response, response to interferon-alpha,

LPS or virus, innate immune response, macrophage activation etc.

I suggest changing to: Enhancers usually engage in chromosomal interactions with their target promoters to regulate gene expression. To explore the genes directly modulated by rs2280381-containing region, we first analyzed the 4C-seq data from the rs2280381 view point, and found that 66 gene promoter regions interact with the rs2280381 site (Supplementary Data Set 2B). Comparing the expression of these genes between WT and rs2280381 KO cell clones, we found that the expression levels of lncRNA AC092723.1 and IRF8 were significantly different between the two groups (\log_2 fold-change ≥ 1 and $FDR < 0.05$) (Supplementary Data Set 2C), suggesting AC092723.1 and IRF8 may be directly regulated by the rs2280381-containing enhancer region. After demonstrating IRF8 as a direct target of rs2280381-containing region, we used the Ingenuity Pathway Analysis (IPA) to predict the possible targets of IRF8. In parallel, IRF8 ChIP-seq data in monocytes (<http://cistrome.org/db/#/>)^{48,49} was analyzed to identify gene promoter region enriched for IRF8 binding, which we considered direct IRF8 targets. These analyses yielded 61 gene candidates (Supplementary Data Set 2D). For some genes, such as ISG15, IFIT1 and CIITA, we performed RT-qPCR to validate their expression. Consistent with the previous reports, IRF8 negatively regulated ISG15 and IFIT1 expression and positively regulated CIITA expression in our data (Fig. S3B-D). Next, differentially expressed genes were analyzed using gene ontology (GO) analysis which revealed that these genes are highly enriched in expected biological process such as inflammatory response, response to interferon-alpha, LPS or virus, innate immune response, and macrophage activation.

Be more precise here please: about 61 gene candidates? Or exactly 61?

Also I would suggest to drop 'etc' after macrophage activation. If there are additional GO categories that the authors feel are worth mentioning, please list them here.

2. Lines 289 – 291: Use of the CRISPR SAM activation system targeting the rs2280381-containing region by gRNA strongly upregulated both AC092723.1 and IRF8 expression (Fig. 3D-E).

I would suggest to change to: Targeting CRISPR SAM to the rs2280381-containing region strongly upregulated both AC092723.1 and IRF8 expression (Fig. 3D-E).

3. Lines 352-354: Consistent with this observation, knockdown AC092723.1 by ASO significantly decreased the interaction intensity between IRF8 promoter and rs2280381 site (Fig. 4G-H),

I suggest to change to: Consistent with this observation, knocking down AC092723.1 by ASO significantly decreased the interaction frequency between the IRF8 promoter and rs2280381 (Fig. 4G-H).

Similarly, I would use 'interaction frequency' instead of 'interaction intensity' in line 398, and also instead of 'connected intensity' in line 624.

4. Line 411: Space missing after IRF8

5. Lines 411-413: (J-K) RT-qPCR analysis of TET1 (I) and IRF8(J) expression in U-937 cells primary monocytes after knockdown of TET1 by siRNA (n = 3, biological replicates). I is crossed out here in the revised version of the manuscript. Is TET1 expression shown in J and IRF8 expression shown in K?

6. Lines 458- 463: To identify the actual proteins binding with rs2280381 site, we first compared the proteins pulling down by rs2280381 sequence and random sequence, the proteins detected only in rs2280381 sequence was considered as the possible candidate proteins. Further, we used the ChIP-seq data of Cistrome database⁴⁸ to collect all the proteins binding with this site (Supplementary Data

Set 3C), the overlapped proteins between DAPA-MS and Cistrome database were selected as our candidate.

I suggest changing to: To identify proteins binding to rs2280381, we first compared the proteins pulled down by the rs2280381 sequence to those pulled down using a random control sequence,. We focused on the proteins detected only using the rs2280381 for the pull down. Further, we used the ChIP-seq data of Cistrome database⁴⁸ to collect all the proteins binding with this site (Supplementary Data Set 3C), and we selected the overlapping proteins between DAPA-MS and Cistrome database as our candidates.

7. Lines 528- 538:...some of which were further confirmed by SAM CRISPR activation assay and KRAB dCas9 interference assay. Usually, enhancers can be mapped by CRISPR mediated deletion, activation and interference^{46,63,64}, and most studies adopted KRAB-dCas9 mediated interference to interfere with enhancer function since KRAB-dCas9 system could induce heterochromatin across about 1 kb window around a gRNA's target site^{46,65-67}. However, in our study, the distance among several SNPs is less than 500 bp. To fine discriminate the function of regulatory elements carrying disease-associated variants, we chose dCas9-VP64 CRISPR activation assay as our priority to perform the screen. Totally, our strategy provides a blueprint for identifying the functional SNPs regulating the expression of genes encoding transcription factors or other molecules.

I suggest changing to:...some of which were further confirmed by SAM CRISPR activation and KRAB dCas9 interference. In general, enhancers can be mapped by CRISPR mediated deletion, activation and interference^{46,63,64}, with KRAB-dCas9 mediated interference of enhancer function arguably the most widely used approach. Notably, the KRAB-dCas9 system has been shown to induce heterochromatin establishment across about 1 kb window around a gRNA's target site^{46,65-67}. However, in our study, the distance between several SNPs and their target genes is less than 500 bp. Therefore, to fine map the function of regulatory elements carrying disease-associated variants, we chose dCas9-VP64 CRISPR activation assay as our priority to perform the screen. We propose that our strategy provides a blueprint for identifying the functional SNPs regulating the expression of genes encoding transcription factors or other key immune response genes.

8. Lines 567 to 582: Genetic studies have identified rs2280381 T allele as the SLE risk allele^{28,31}, our results show that rs2280381 risk allele T results in lower expression of IRF8 than the non-risk allele C in monocytes, this is consistent with the eQTL data in primary monocytes from ImmuneXUT database⁷⁴, which means that low expression of IRF8 in monocytes is the risk factor for SLE. It seems controversial with the report that IRF8 deficient reduced the autoimmunity in NZB mice since lack of plasmacytoid dendritic cells⁷⁵. However, the role of IRF8 in autoimmune is paradoxical, conditional KO IRF8 in B cells results in the breakdown of B cell tolerance thus contributing to the autoimmune disease development⁷⁶. Consistent with this finding, our data and public RNA profiling data reveal the down-regulated expression of IRF8 in SLE patients (Fig. S5F-I). Moreover, IRF8 impacts the differentiation of DCs, monocytes and neutrophils are dose-dependent. The low dose of IRF8 could induce the differentiation of pDCs and patrolling monocytes⁷⁷, which are the cell types promoting the pathogenesis of lupus^{75,78}. Together, the role of IRF8 in SLE is very complex. It is likely that the function of IRF8 in SLE is stage-specific or cell-type-specific, which still need to be carefully deciphered in the future.

Several phrases in the above paragraph are not clear to me. For example: It seems controversial with the report that IRF8 deficient reduced the autoimmunity in NZB mice since lack of plasmacytoid dendritic cells⁷⁵. This is an example of a paragraph that would benefit from professional editing.

9. Lines 609-615: Notably, we also observed other proteins binding with rs2280381 site in the DAPA

assay (Supplementary Data set 3A-B), and the maintenance of cell-type-specific enhancer was mediated by a series of LDTFs and chromatin regulators, so whether other proteins also mediated the allele-specific regulation of rs2280381 on IRF8 expression, and the mechanism directed the cell-type specific enhancer activation in this site still deserve to be studied in more detail and depth.

I suggest changing to: Notably, our DAPA experiments also revealed binding of several other proteins to rs2280381 (Supplementary Data set 3A-B), including LDTFs and chromatin regulators. Whether these factors contribute to the allele-specific regulation of rs2280381 on IRF8 expression, and if so, which mechanism they use to direct the cell-type specific enhancer activation at this site still will be the subject of future studies.

REVIEWERS' COMMENTS

Reviewer #1 (Remarks to the Author):

Author addressed all of critiques raised in the previous review.

Response: Thanks for your positive evaluation of our manuscript and for the insightful comments and constructive suggestions to help us improve the manuscript.

Reviewer #2 (Remarks to the Author):

Zhou and colleagues have made extensive revisions to address the concerns raised in their original manuscript. They performed a suite of new experiments including ChIRP-seq, CRISPR-SAM, 4C assay, bisulfite-seq, and luciferase assays. Additionally, they conducted new analysis of their own data and publicly available data to address reviewers' questions. The current manuscript is strengthened significantly. I only have a minor suggestion. The luciferase reporter assay results in Fig. R24 are complementary to the data shown in Fig. 5. I feel it should be included in that figure. Otherwise, I congratulate the authors for a great study.

Response: Thanks for your positive evaluation of our manuscript and insightful suggestions. We have added this figure as supplementary figure in the revised manuscript. Your constructive comments strongly help us improved the quality of manuscript.

Reviewer #3 (Remarks to the Author):

The authors have done a really good job to answer my questions and importantly, they have included additional experimental data (CRISPRi) which have improved the quality of the manuscript and further strengthen their conclusions. I recommend this manuscript for publication; however, having gone through the changes in the manuscript, there are many passages where the quality of the data is not matched by the language used. This is a shame and I would urge the authors to enlist the help from a professional editor to bring the writing up to the level that their work deserves to be described in. As a non-native English speaker, I can completely relate to the difficulties the authors are experiencing here. Having faced similar issues, I have used professional editors in the past and it has definitely helped to improve the quality of my manuscripts. I strongly recommend the authors do the same for this manuscript.

Response: Thanks for your insightful suggestion. We deeply appreciate your constructive comments that greatly help improve the presentation of this manuscript. In addition, we have revised the manuscript with the help of professional native-English speakers.

I am outlining a few examples below where I feel the writing could be improved; however, there are more and the entire manuscript would benefit from an overall review by a

professional.

1. Lines 253-275: Enhancers usually form gene loop with the promoter site to regulate target gene expression. To explore the genes directly modulated by rs2280381-containing region, we first analyzed the 4C-seq data from the rs2280381 view point and find that 66 genes' promoter regions exist interaction with the rs2280381 site (Supplementary Data Set 2B). Analysis of these genes' expression between WT and rs2280381 KO cell clones, we found that the expressions of lncRNA AC092723.1 and IRF8 were significantly different between the two groups (\log_2 fold-change ≥ 1 and $FDR < 0.05$) (Supplementary Data Set 2C), suggesting AC092723.1 and IRF8 may be directly regulated by the rs2280381-containing enhancer region. After demonstrating IRF8 as a direct target of rs2280381-containing region, we adopted the Ingenuity Pathway Analysis (IPA) to predict the possible targets of IRF8. Meanwhile, IRF8 ChIP-seq data in monocytes (<http://cistrome.org/db/>)48,49 was analyzed to find whether IRF8 could bind to the promoter region of predicted genes, the genes' promoter region enriched with IRF8 binding were considered as IRF8 direct target. Finally, we got about 61 gene candidates (Supplementary Data Set 2D). For some genes, such as ISG15, IFIT1 and CIITA, we performed RT-qPCR to validate their expression. Consistent with the previous reports, IRF8 could negatively regulated ISG1550 and IFIT136 expression and positively regulated CIITA51 expression in our data (Fig. S3B-D). Next, differentially expressed genes were analyzed using gene ontology (GO) analysis and revealed that these genes are highly enriched in expected biological process such as inflammatory response, response to interferon-alpha, LPS or virus, innate immune response, macrophage activation etc.

I suggest changing to: Enhancers usually engage in chromosomal interactions with their target promoters to regulate gene expression. To explore the genes directly modulated by rs2280381-containing region, we first analyzed the 4C-seq data from the rs2280381 view point, and found that 66 gene promoter regions interact with the rs2280381 site (Supplementary Data Set 2B). Comparing the expression of these genes between WT and rs2280381 KO cell clones, we found that the expression levels of lncRNA AC092723.1 and IRF8 were significantly different between the two groups (\log_2 fold-change ≥ 1 and $FDR < 0.05$) (Supplementary Data Set 2C), suggesting AC092723.1 and IRF8 may be directly regulated by the rs2280381-containing enhancer region. After demonstrating IRF8 as a direct target of rs2280381-containing region, we used the Ingenuity Pathway Analysis (IPA) to predict the possible targets of IRF8. In parallel, IRF8 ChIP-seq data in monocytes (<http://cistrome.org/db/>)48,49 was analyzed to identify gene promoter region enriched for IRF8 binding, which we considered direct IRF8 targets. These analyses yielded 61 gene candidates (Supplementary Data Set 2D). For some genes, such as ISG15, IFIT1 and CIITA, we performed RT-qPCR to validate their expression. Consistent with the previous reports, IRF8 negatively regulated ISG1550 and IFIT136 expression and positively regulated CIITA51 expression in our data (Fig. S3B-D). Next, differentially expressed genes were analyzed using gene ontology (GO) analysis which revealed that these genes are highly enriched in expected biological process such as inflammatory response, response to interferon-alpha, LPS or virus, innate immune response, and macrophage

activation.

Be more precise here please: about 61 gene candidates? Or exactly 61?

Also I would suggest to drop 'etc' after macrophage activation. If there are additional GO categories that the authors feel are worth mentioning, please list them here.

Response: Thanks for your insightful suggestion. We have revised these sentences in the manuscript.

2. Lines 289 – 291: Use of the CRISPR SAM activation system targeting the rs2280381-containing region by gRNA strongly upregulated both AC092723.1 and IRF8 expression (Fig. 3D-E).

I would suggest to change to: Targeting CRISPR SAM to the rs2280381-containing region strongly upregulated both AC092723.1 and IRF8 expression (Fig. 3D-E).

Response: Thanks for your insightful suggestion. We have revised this sentence in the manuscript.

3. Lines 352-354: Consistent with this observation, knockdown AC092723.1 by ASO significantly decreased the interaction intensity between IRF8 promoter and rs2280381 site (Fig. 4G-H),

I suggest to change to: Consistent with this observation, knocking down AC092723.1 by ASO significantly decreased the interaction frequency between the IRF8 promoter and rs2280381 (Fig. 4G-H).

Similarly, I would use 'interaction frequency' instead of 'interaction intensity' in line 398, and also instead of 'connected intensity' in line 624.

Response: Thanks for your insightful suggestion. We have revised these words in the manuscript.

4. Line 411: Space missing after IRF8

Response: We are sorry for this mistake, and we have revised this in the manuscript.

5. Lines 411-413: (J-K) RT-qPCR analysis of TET1 (I) and IRF8(J) expression in U-937 cells primary monocytes after knockdown of TET1 by siRNA (n = 3, biological replicates). I is crossed out here in the revised version of the manuscript. Is TET1 expression shown in J and IRF8 expression shown in K?

Response: We are sorry for this mistake, and we have revised this in the manuscript.

6. Lines 458- 463: To identify the actual proteins binding with rs2280381 site, we first compared the proteins pulling down by rs2280381 sequence and random sequence, the proteins detected only in rs2280381 sequence was considered as the possible candidate proteins. Further, we used the CHIP-seq data of Cistrome database⁴⁸ to collect all the proteins binding with this site (Supplementary Data Set 3C), the overlapped proteins between DAPA-MS and Cistrome database were selected as our candidate.

I suggest changing to: To identify proteins binding to rs2280381, we first compared the proteins pulled down by the rs2280381 sequence to those pulled down using a random control sequence. We focused on the proteins detected only using the rs2280381 for the pull down. Further, we used the CHIP-seq data of Cistrome database⁴⁸ to collect all the proteins binding with this site (Supplementary Data Set 3C), and we selected the overlapping proteins between DAPA-MS and Cistrome database as our candidates.

Response: Thanks for your insightful suggestion. We have revised these sentences in the manuscript.

7. Lines 528- 538:...some of which were further confirmed by SAM CRISPR activation assay and KRAB dCas9 interference assay. Usually, enhancers can be mapped by CRISPR mediated deletion, activation and interference^{46,63,64}, and most studies adopted KRAB-dCas9 mediated interference to interfere with enhancer function since KRAB-dCas9 system could induce heterochromatin across about 1 kb window around a gRNA's target site^{46,65-67}. However, in our study, the distance among several SNPs is less than 500 bp. To fine discriminate the function of regulatory elements carrying disease-associated variants, we chose dCas9-VP64 CRISPR activation assay as our priority to perform the screen. Totally, our strategy provides a blueprint for identifying the functional SNPs regulating the expression of genes encoding transcription factors or other molecules.

I suggest changing to:...some of which were further confirmed by SAM CRISPR activation and KRAB dCas9 interference. In general, enhancers can be mapped by CRISPR mediated deletion, activation and interference^{46,63,64}, with KRAB-dCas9 mediated interference of enhancer function arguably the most widely used approach. Notably, the KRAB-dCas9 system has been shown to induce heterochromatin establishment across about 1 kb window around a gRNA's target site^{46,65-67}. However, in our study, the distance between several SNPs and their target genes is less than 500 bp. Therefore, to fine map the function of regulatory elements carrying disease-associated variants, we chose dCas9-VP64 CRISPR activation assay as our priority to perform the screen. We propose that our strategy provides a blueprint for identifying the functional SNPs regulating the expression of genes encoding transcription factors or other key immune response genes.

Response: Thanks for your insightful suggestion. We have revised these sentences in the manuscript.

8. Lines 567 to 582: Genetic studies have identified rs2280381 T allele as the SLE risk allele^{28,31}, our results show that rs2280381 risk allele T results in lower expression of IRF8 than the non-risk allele C in monocytes, this is consistent with the eQTL data in primary monocytes from ImmuneNexUT database⁷⁴, which means that low expression of IRF8 in monocytes is the risk factor for SLE. It seems controversial with the report that IRF8 deficient reduced the autoimmunity in NZB mice since lack of plasmacytoid dendritic cells⁷⁵. However, the role of IRF8 in autoimmune is paradoxical, conditional KO IRF8 in B cells results in the breakdown of B cell tolerance thus contributing to the autoimmune disease development⁷⁶. Consistent with this finding, our data and public RNA profiling data reveal the down-regulated expression of IRF8 in SLE patients (Fig. S5F-I). Moreover, IRF8 impacts the differentiation of DCs, monocytes and neutrophils are dose-dependent. The low dose of IRF8 could induce the differentiation of pDCs and patrolling monocytes⁷⁷, which are the cell types promoting the pathogenesis of lupus^{75,78}. Together, the role of IRF8 in SLE is very complex. It is likely that the function of IRF8 in SLE is stage-specific or cell-type-specific, which still need to be carefully deciphered in the future.

Several phrases in the above paragraph are not clear to me. For example: It seems controversial with the report that IRF8 deficient reduced the autoimmunity in NZB mice since lack of plasmacytoid dendritic cells⁷⁵. This is an example of a paragraph that would benefit from professional editing.

Response: Thanks for pointing out this. We have re-written these sentences in the revised manuscript.

9. Lines 609-615: Notably, we also observed other proteins binding with rs2280381 site in the DAPA assay (Supplementary Data set 3A-B), and the maintenance of cell-type-specific enhancer was mediated by a series of LDTFs and chromatin regulators, so whether other proteins also mediated the allele-specific regulation of rs2280381 on IRF8 expression, and the mechanism directed the cell-type specific enhancer activation in this site still deserve to be studied in more detail and depth.

I suggest changing to: Notably, our DAPA experiments also revealed binding of several other proteins to rs2280381 (Supplementary Data set 3A-B), including LDTFs and chromatin regulators. Whether these factors contribute to the allele-specific regulation of rs2280381 on IRF8 expression, and if so, which mechanism they use to direct the cell-type specific enhancer activation at this site still will be the subject of future studies.

Response: Thanks for your insightful suggestion. We have revised these sentences in the manuscript.